# Meta-Reinforcement Learning with Adaptation from Human Feedback via Preference-Order-Preserving Task Embedding

Siyuan Xu [1]    Minghui Zhu [1]

## Abstract

This paper studies meta-reinforcement learning with adaptation from human feedback. It aims to pre-train a meta-model that can achieve few-shot adaptation for new tasks from human preference queries without relying on reward signals. To solve the problem, we propose the framework *adaptation via Preference-Order-preserving EMbedding* (POEM). In the meta-training, the framework learns a task encoder, which maps tasks to a preference-order-preserving task embedding space, and a decoder, which maps the embeddings to the task-specific policies. In the adaptation from human feedback, the task encoder facilitates efficient task embedding inference for new tasks from the preference queries and then obtains the task-specific policy. We provide a theoretical guarantee for the convergence of the adaptation process to the task-specific optimal policy and experimentally demonstrate its state-of-the-art performance with substantial improvement over baseline methods.

## 1. Introduction

Reinforcement learning (RL) has achieved significant success in various sequential decision-making tasks, including video games (Mnih et al., 2015; Silver et al., 2016; Lee et al., 2018), robotics (Levine et al., 2016; Lee et al., 2020; Margolis et al., 2021; 2024), and quantitative finance (Liu et al., 2021). However, when applied to real-world problems, conventional RL methods encounter the difficulty of designing a proper reward function (Kaufmann et al., 2023). In RL, an agent seeks to identify the optimal policy that maximizes an accumulative reward function using reward signals. The reward function needs to be properly chosen such that the agent's state-action pairs align well with the task's goal. However, designing such a reward function can be difficult, especially for tasks where the goals are too ambiguous to be directly mapped to specific state-action pairs (Christiano et al., 2017; Bai et al., 2022), such as a robot cleaning a room or preparing a meal. Moreover, agents may be required to offer personalized services tailored to human users. Due to lack of domain knowledge, it is hard for the human users to design a reward function that reflects their objective. To address the challenge, RL from human feedback (RLHF) (Casper et al., 2023; Kaufmann et al., 2023) aims to solve RL using human feedback, e.g., the binary trajectory comparisons or trajectory rankings, instead of reward signals. Specifically, RLHF learns a reward function by aligning the reward signals along trajectories with human preferences and then optimizes the policy by off-the-shelf RL algorithms under the learned reward function. RLHF has achieved state-of-the-art results in large language model (LLM) fine-tuning (Ouyang et al., 2022) and preference optimization of text-to-image generation (Lee et al., 2023).

Although it addresses the challenge of reward design, RLHF faces the challenge of data inefficiency. First, the large state-action space necessitates substantial preference data to avoid overfitting the learned reward function. However, due to the limitation in annotators' expertise, capabilities, and attention, collecting high-quality human feedback data incurs significant costs and takes considerable time (Casper et al., 2023). Furthermore, optimizing the policy in RLHF relies on conventional RL algorithms, which require extensive environment exploration, often amounting to millions of state transition data.

Meta-reinforcement learning (meta-RL) (Finn et al., 2017; Beck et al., 2023) aims to acquire transferable knowledge by identifying common structures across multiple prior tasks during meta-training. At meta-test time, this knowledge enables the adaptation to unseen tasks using a small amount of data, thereby accelerating the training process and improving the data efficiency of RL algorithms. Incorporating meta-RL into human-in-the-loop adaptation (Ren et al., 2022; Joey Hejna, 2023) facilitates few-shot adaptation from preference queries without relying on reward signals, and

[1]School of Electrical Engineering and Computer Science, Pennsylvania State University, University Park, USA. Correspondence to: Minghui Zhu <muz16@psu.edu>, Siyuan Xu <spx5032@psu.edu>.

*Proceedings of the 42nd International Conference on Machine Learning*, Vancouver, Canada. PMLR 267, 2025. Copyright 2025 by the author(s).

*Table 1.* Available data for meta-RL and meta-RL with adaptation from human feedback

|  | Meta-RL | Meta-RL with adaptation from human feedback |
|---|---|---|
| Meta-training | Trajectories with rewards | Trajectories with rewards |
| Meta-test | Few trajectories with rewards | Few trajectories, few preference queries of trajectories |

therefore offers a promising solution to the data inefficiency problem in RLHF. Paper (Joey Hejna, 2023) applies a supervised meta-learning approach, MAML (Finn et al., 2017), to train a meta-reward model, and adapts it to a task-specific reward model with few-shot human preference data during the meta-test. However, the meta-test still needs to solve the RL problem to obtain the policy under the learned task-specific reward model, which requires a large amount of state transition data. ANOLE (Ren et al., 2022) adopts the context-based meta-RL method (Rakelly et al., 2019). During the meta-training, ANOLE uses reward signals to learn a task encoder, which maps tasks to the task embeddings, and a decoder, which maps the task embeddings to the task-specific policies. The meta-test infers the task embedding of a new task from its human preference data and then derives its policy. Notice that ANOLE uses the reward signals for the meta-training and the preference data for the meta-test. The mismatch of data types prevents the task encoder from capturing preference-related information across tasks, and therefore degrades the performance of task-specific policies.

**Main contribution.** In this paper, we propose the framework *adaptation via Preference-Order-preserving EMbedding* (POEM) for meta-RL with adaptation from human feedback. During meta-training, a task encoder is trained with preference data to ensure that the task embedding space maintains the preference-order-preserving property, i.e., the optimal policy for one task is preferred under another task if the embeddings of the two tasks have high similarity. During the adaptation from human feedback, when a new task is given by a human, all task embeddings that do not align with the human's preference orders are rejected. Among all the remaining task embeddings, the one with the lowest preference loss on the preference order data is selected to derive the task-specific policy. Our contributions are three-fold. From the algorithm perspective, this paper is the first to propose the preference-order-preserving task encoder for context-based meta-RL training, which establishes a connection between task embeddings and human preferences. This connection facilitates the efficient inference of task embeddings for new tasks during the adaptation from human feedback. From the experiment perspective, we conduct experiments on nine continuous control environments in Mujoco and MetaWorld. The proposed POEM, using few-shot human preference queries, achieves a performance comparable to meta-RL oracle (using reward signals) and demonstrates state-of-the-art results with 20%-50% per-

formance improvement. From the theory perspective, we derive a theoretical result for POEM based on the preference-order-preserving property of the embedding space, which guarantees the convergence of the policy distribution generated by POEM to the optimal task-specific policy.

**Related works.** Due to the space limit, related works are included in Appendix A.

## 2. Problem Statement

**MDP and RL task.** A Markov decision process (MDP) $\mathcal{M} \triangleq \{\mathcal{S}, \mathcal{A}, \gamma, \rho, P, r\}$ is defined by the state space $\mathcal{S}$, the action space $\mathcal{A}$, the discount factor $\gamma$, the initial state distribution $\rho$ over $\mathcal{S}$, the transition probability $P(s'|s, a)$ $: \mathcal{S} \times \mathcal{A} \times \mathcal{S} \to [0, 1]$, and the reward function $r : \mathcal{S} \times \mathcal{A} \to [0, r_{max}]$. A stochastic policy $\pi : \mathcal{S} \to \mathbb{P}(\mathcal{A})$ is a map from states to probability distributions over actions, and $\pi(a|s)$ denotes the probability of selecting action $a$ in state $s$. The accumulated reward function is defined as $J(\pi) \triangleq \mathbb{E}_{s_0 \sim \rho}[\sum_{t=0}^{\infty} \gamma^t r(s_t, a_t, s_{t+1})|\pi]$. An RL task is to identify the optimal policy that can maximize the accumulated reward function i.e., $\pi^* \triangleq \arg\max_{\pi} J(\pi)$.

**RL task distribution.** Consider a space of RL tasks $\Gamma$, where each task $\mathcal{T} \in \Gamma$ is modeled by an MDP $\mathcal{M}_{\mathcal{T}} \triangleq \{\mathcal{S}, \mathcal{A}, \gamma, \rho, P_{\mathcal{T}}, r_{\mathcal{T}}\}$, where $P_{\mathcal{T}}$ and $r_{\mathcal{T}}$ condition on the task $\mathcal{T}$. For task $\mathcal{T}$, the accumulated reward function is denoted as $J_{\mathcal{T}}(\pi)$, and the optimal policy is denoted as $\pi_{\mathcal{T}}^*$. Assume the RL tasks follow a task probability distribution $\mathbb{P}(\Gamma)$.

**Meta-RL with adaptation from human feedback** aims to learn a meta-model from the task distribution $\mathbb{P}(\Gamma)$, which can be adapted to an unseen task $\mathcal{T}_{new} \sim \mathbb{P}(\Gamma)$ using few-shot human-preference data and few-shot state transition data without the reward signals.

During the meta-training, a set of tasks $\{\mathcal{T}_j\}_{j=1}^N$ are sampled from $\mathbb{P}(\Gamma)$, and the tasks' MDPs $\{\mathcal{M}_{\mathcal{T}_j}\}_{j=1}^N$ are explored to generate trajectories. Note that, the reward function $r_{\mathcal{T}_j}$ of each meta-training task $\mathcal{T}_j$ is accessible. Consequently, trajectories with rewards defined as $\{(s_t, a_t, s_{t+1}, r_t)\}_{t=0}^H$ can be obtained for each task $\mathcal{T}_j$.

During the meta-test, a new task $\mathcal{T}_{new}$ is given by a human, and the MDP $\mathcal{M}_{\mathcal{T}_{new}}$ can be explored by the agent to obtain trajectories $\tau = \{(s_t, a_t, s_{t+1})\}_{t=0}^H$. However, the rewards along the trajectories cannot be returned. The preference order of a pair of trajectories $(\tau_1, \tau_2)$ can be queried. For

any given query trajectory pair $(\tau_1, \tau_2)$, the preference oracle (human) would return $\tau_1 \succ_{\mathcal{T}_{new}} \tau_2$ if the oracle prefers $\tau_1$ over $\tau_2$ under $\mathcal{T}$; otherwise, $\tau_1 \prec_{\mathcal{T}_{new}} \tau_2$ is returned. If the preference oracle does not distinguish $\tau_1$ and $\tau_2$, any of $\tau_1 \succ_{\mathcal{T}_{new}} \tau_2$ and $\tau_1 \prec_{\mathcal{T}_{new}} \tau_2$ can be returned. Note that, during meta-testing, both the numbers of sampled trajectories and human preference queries are small. A comparison of available data in conventional meta-RL (Finn et al., 2017; Beck et al., 2023) and meta-RL with adaptation from human feedback is shown in Table 1.

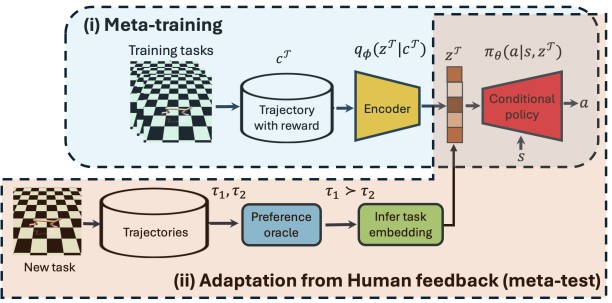

*Figure 1.* Overview of meta-RL with adaptation from human feedback via preference-order-preserving embedding (POEM)

## 3. Method Overview

This section presents an overview of our method, *adaptation via Preference-Order-preserving EMbedding* (POEM), to solve the meta-RL with adaptation from human feedback. We introduce **the preference-order-preserving embedding** and design two modules for POEM: **(i) meta-training with preference-order-preserving embedding** and **(ii) adaptation from human feedback by task embedding inference**. The structure of the two modules is shown in Figure 1.

The preference-order-preserving embedding is to encode each task $\mathcal{T}$ to a task embedding $z^{\mathcal{T}}$, where the embeddings adhere to the preference-order-preserving property. The preference-order-preserving property refers to that, if the optimal policy of a task $\mathcal{T}_1$, denoted as $\pi^*_{\mathcal{T}_1}$, is preferred over that of another task $\mathcal{T}_2$, denoted as $\pi^*_{\mathcal{T}_2}$, under a task $\mathcal{T}_0$, this preference order should be reflected by the similarity order of the task embeddings, i.e., the similarity (inverse measure of vector distance) between the task embedding vectors $z^{\mathcal{T}_0}$ and $z^{\mathcal{T}_1}$, denoted as $S(z^{\mathcal{T}_0}, z^{\mathcal{T}_1})$, is higher than the similarity $S(z^{\mathcal{T}_0}, z^{\mathcal{T}_2})$ between $z^{\mathcal{T}_0}$ and $z^{\mathcal{T}_2}$. Formally, $S(z^{\mathcal{T}_0}, z^{\mathcal{T}_1}) \geq S(z^{\mathcal{T}_0}, z^{\mathcal{T}_2})$ holds if $\tau^{\mathcal{T}_0}(\pi^*_{\mathcal{T}_1}) \succ_{\mathcal{T}_0} \tau^{\mathcal{T}_0}(\pi^*_{\mathcal{T}_2})$, where $\tau^{\mathcal{T}_0}(\pi)$ denotes the trajectory sampled by policy $\pi$ on task $\mathcal{T}_0$. The property is further illustrated by Figure 2. With the preference-order-preserving property, the preference order of the task-specific optimal policies can be directly derived from the task embeddings. More importantly, when a task is unknown but the preference order under the task is available from a human, the information

about the task embeddings can be inferred.

In the module of meta-training, we train a preference-order-preserving task encoder and a policy decoder. Specifically, the encoder $q_\phi(z^{\mathcal{T}}|c^{\mathcal{T}})$ encodes the task-specific trajectories with rewards (referred to as context $c^{\mathcal{T}}$) into a task embedding vector $z^{\mathcal{T}}$ for each training task $\mathcal{T}$, and the decoder policy $\pi_\theta(a|s, z^{\mathcal{T}})$ conditions on the task embedding $z^{\mathcal{T}}$ to obtain the task-specific policy for task $\mathcal{T}$. Two goals are expected for training $q_\phi$ and $\pi_\theta$: the preference-order-preserving property holds for the encoder $q_\phi$ and the conditional policy $\pi_\theta(, z^{\mathcal{T}})$ is close to the optimal policy of task $\mathcal{T}$. We incorporate both the preference data and the reward signals to the meta-training to achieve the two goals.

In the module of adaptation from human feedback, for a given new task $\mathcal{T}_{new}$, we sample the trajectory pairs $(\tau_1, \tau_2)$, query the preference oracle to generate the preference order of $(\tau_1, \tau_2)$, and then infer the task embedding $z^{\mathcal{T}_{new}}$ to obtain the task-specific policy $\pi_\theta(\cdot|\cdot, z^{\mathcal{T}_{new}})$. In particular, for the task embedding inference, we first sample a set of task embedding candidates. Based on the preference-order-preserving property, we identify the preference order of the trajectories $(\tau_1, \tau_2)$ under each embedding candidate, and remove all embeddings that do not align with the queried preference orders on the new task $\mathcal{T}_{new}$. Among all the remaining task embeddings, the one with the lowest preference loss on the preference query data is selected to derive the task-specific policy.

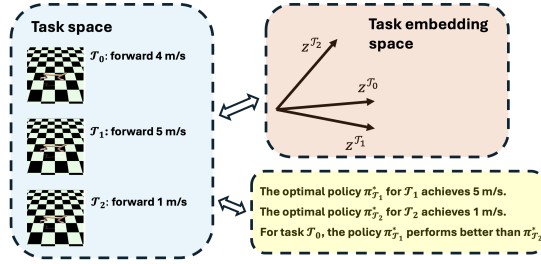

*Figure 2.* An example to illustrate the insights of the preference-order-preserving task embedding. Consider three RL tasks $\mathcal{T}_0, \mathcal{T}_1$ and $\mathcal{T}_2$ as steering the robot moving forward with $4\ m/s$, $5\ m/s$, and $1\ m/s$, respectively. The task embedding $z^{\mathcal{T}_0}$ is more similar to $z^{\mathcal{T}_1}$ than $z^{\mathcal{T}_2}$, and $\pi^*_{\mathcal{T}_1}$ performs better than $\pi^*_{\mathcal{T}_2}$ on task $\mathcal{T}_0$.

Note that the encoder-decoder structure in the meta-training of Figure 1 is inspired by the context-based meta-RL methods (Rakelly et al., 2019; Zintgraf et al., 2020; Raileanu et al., 2020). As shown in Table 1, the task context (trajectories with rewards) can be obtained for both meta-training and meta-test tasks in meta-RL. Then, in both meta-training and adaptation (meta-test) of context-based meta-RL methods, the encoder-decoder structure (the blue box) in Figure 1 is applied to encode tasks and decode task-specific policies. However, for meta-RL with adaptation from human feed-

back, reward signals are not available and only preference data can be accessed during the meta-test. ANOLE (Ren et al., 2022) simply borrows the encoder-decoder from the context-based meta-RL methods and resorts to training an extra reward decoder which generates reward signals during the meta-test. In particular, during the meta-training, the task encoder maps the task-specific trajectory with reward signals to a task embedding, and the task embedding is decoded to both the task-specific policy and reward function by a policy decoder and a reward decoder. However, the task encoder cannot capture preference-related features across tasks, and therefore degrades the performance. In contrast, we design the novel task encoder, which extracts task features incorporating preference order information, enabling efficient inference of task embeddings from human preference.

## 4. Preference-Order-Preserving Embedding

As mentioned in Section 3, the preference-order-preserving task embedding connects the relations over the task embeddings with human preference feedback. The design is motivated by the following two insights.

**Insight 1.** The similarity between tasks should be reflected in the similarity of their embeddings. In specific, if task $\mathcal{T}_0$ is more similar to $\mathcal{T}_1$ than $\mathcal{T}_2$, the similarity between the embedding vectors of $\mathcal{T}_0$ and $\mathcal{T}_1$, denoted as $S(z^{\mathcal{T}_0}, z^{\mathcal{T}_1})$, should be higher than $S(z^{\mathcal{T}_0}, z^{\mathcal{T}_2})$, and vice versa. Formally,

$$\mathcal{T}_0 \text{ is more (or equally) similar to } \mathcal{T}_1 \text{ than to } \mathcal{T}_2$$
$$\iff S(z^{\mathcal{T}_0}, z^{\mathcal{T}_1}) \geq S(z^{\mathcal{T}_0}, z^{\mathcal{T}_2}) .$$

This insight is illustrated on the top side of Figure 2 and is widely applied in the design of representation learning methods for images and natural language, such as contrastive learning (Chen et al., 2020; Radford et al., 2021).

**Insight 2.** The second insight comes from an intuition of task similarity: if two tasks are similar, the optimal policy of one task can also perform relatively well on the other task. In specific, consider the optimal policies $\pi_{\mathcal{T}_1}^*$ of $\mathcal{T}_1$ and $\pi_{\mathcal{T}_2}^*$ of $\mathcal{T}_2$. If $\mathcal{T}_0$ is more similar to $\mathcal{T}_1$ than to $\mathcal{T}_2$, the optimal policy $\pi_{\mathcal{T}_0}^*$ for task $\mathcal{T}_0$ should be closer to $\pi_{\mathcal{T}_1}^*$ than to $\pi_{\mathcal{T}_2}^*$, then $\pi_{\mathcal{T}_1}^*$ should achieve better performance on $\mathcal{T}_0$ than $\pi_{\mathcal{T}_2}^*$, i.e., the total reward $J_{\mathcal{T}_0}(\pi_{\mathcal{T}_1}^*) > J_{\mathcal{T}_0}(\pi_{\mathcal{T}_2}^*)$. Formally,

$$\mathcal{T}_0 \text{ is more (or equally) similar to } \mathcal{T}_1 \text{ than to } \mathcal{T}_2$$
$$\iff J_{\mathcal{T}_0}(\pi_{\mathcal{T}_1}^*) \geq J_{\mathcal{T}_0}(\pi_{\mathcal{T}_2}^*) .$$

This insight is illustrated at the bottom of Figure 2.

Starting from Insights 1 and 2, we can derive the property for the preference-order-preserving embedding. Specifically, combining Insights 1 and 2, for any tasks $\mathcal{T}_0$, $\mathcal{T}_1$, and $\mathcal{T}_2$,

$$S(z^{\mathcal{T}_0}, z^{\mathcal{T}_1}) \geq S(z^{\mathcal{T}_0}, z^{\mathcal{T}_2}) \iff J_{\mathcal{T}_0}(\pi_{\mathcal{T}_1}^*) \geq J_{\mathcal{T}_0}(\pi_{\mathcal{T}_2}^*). \quad (1)$$

Next, following most existing works of RLHF (Wirth et al., 2017; Kaufmann et al., 2023), the preference order between two trajectories $\tau_1$ and $\tau_2$ is modeled as the order of the received (discounted) total rewards, i.e., $\tau_1 \succ_{\mathcal{T}} \tau_2$ implies the total reward $\sum_{t=0}^{H} \gamma^t r_{\mathcal{T}}(s_t^{(1)}, a_t^{(1)}, s_{t+1}^{(1)})$ received from $\tau_1$ is larger than (or equal to) $\sum_{t=0}^{H} \gamma^t r_{\mathcal{T}}(s_t^{(2)}, a_t^{(2)}, s_{t+1}^{(2)})$ received from $\tau_2$. Then, we approximate the expected total reward $J_{\mathcal{T}_0}(\pi_{\mathcal{T}}^*)$ in (1) by the average total reward of multiple sampled trajectories generated from $\pi_{\mathcal{T}}^*$ under $\mathcal{T}_0$, i.e., approximately we have

$$J_{\mathcal{T}_0}(\pi_{\mathcal{T}_1}^*) \geq J_{\mathcal{T}_0}(\pi_{\mathcal{T}_2}^*) \iff \tau^{\mathcal{T}_0}(\pi_{\mathcal{T}_1}^*) \succ_{\mathcal{T}_0} \tau^{\mathcal{T}_0}(\pi_{\mathcal{T}_2}^*), \quad (2)$$

where $\tau^{\mathcal{T}_0}(\pi_{\mathcal{T}_1}^*)$ is the concatenation of multiple trajectories generated by $\pi_{\mathcal{T}_1}^*$, and $\tau^{\mathcal{T}_0}(\pi_{\mathcal{T}_2}^*)$ is that generated by $\pi_{\mathcal{T}_2}^*$. Replacing the right-hand side of (1) by that of (2), we obtain the following preference-order-preserving property.

**Property 1.** *For any tasks $\mathcal{T}_0$, $\mathcal{T}_1$, $\mathcal{T}_2$ with their embeddings $z^{\mathcal{T}_0}$, $z^{\mathcal{T}_1}$, $z^{\mathcal{T}_2}$, we have*

$$S(z^{\mathcal{T}_0}, z^{\mathcal{T}_1}) \geq S(z^{\mathcal{T}_0}, z^{\mathcal{T}_2}) \iff \tau^{\mathcal{T}_0}(\pi_{\mathcal{T}_1}^*) \succ_{\mathcal{T}_0} \tau^{\mathcal{T}_0}(\pi_{\mathcal{T}_2}^*).$$

From Property 1, the similarity ordering of task embedding pairs is expected to align with human preference order, i.e., a task with a more similar embedding is preferred. Leveraging this property, the preference-order-preserving task embedding space establishes a clear relationship between task embeddings and human preference feedback.

Finally, we select the similarity metric $S$ used for Property 1. A common selection of the similarity metric between two embeddings is their inner product (cosine similarity), i.e., $S(z^{\mathcal{T}_1}, z^{\mathcal{T}_2}) \triangleq \langle z^{\mathcal{T}_1}, z^{\mathcal{T}_2} \rangle$. However, if we adopt this similarity metric in this problem, we can prove that it is possible that a task embedding space that satisfies (1) does not exist. The formal statement is shown as Proposition 1 in Appendix B.1.1. In contrast, in the following theorem, we provide a similarity metric that guarantees the existence of the task embedding space that adheres to (1). Then, Property 1 holds when using the approximation stated in (2).

**Theorem 1.** *For any task space $\Gamma$, there exists task encoder mappings $f_r$ and $f_\pi : \Gamma \to \mathbb{R}^d$, such that for any $\mathcal{T}_0$, $\mathcal{T}_1$ and $\mathcal{T}_2 \in \Gamma$, the property in (1) holds when $S(z^{\mathcal{T}_i}, z^{\mathcal{T}_j}) = \langle z_r^{\mathcal{T}_i}, z_\pi^{\mathcal{T}_j} \rangle$, $z_r^{\mathcal{T}_i} = f_r(\mathcal{T}_i)$ and $z_\pi^{\mathcal{T}_i} = f_\pi(\mathcal{T}_i)$ for $i, j = 0, 1$ and $2$. In addition, for any task $\mathcal{T} \in \Gamma$, $f_r(\mathcal{T})$ only depends on $r_{\mathcal{T}}$ and $f_\pi(\mathcal{T})$ only depends on $\pi_{\mathcal{T}}^*$ and $P_{\mathcal{T}}$.*

The proof of Theorem 1 is shown in Appendix B.1.2. As stated in Theorem 1, the task embedding vector $z^{\mathcal{T}}$ can be separated to two vectors: $z^{\mathcal{T}} \triangleq [z_\pi^{\mathcal{T}}, z_r^{\mathcal{T}}]$ and the similarity between $z^{\mathcal{T}_1}$ and $z^{\mathcal{T}_2}$ is defined as

$$S(z^{\mathcal{T}_1}, z^{\mathcal{T}_2}) \triangleq \langle z_r^{\mathcal{T}_1}, z_\pi^{\mathcal{T}_2} \rangle. \quad (3)$$

As indicated in Theorem 1, the embedding $z_r^{\mathcal{T}}$ represents the features of the task-specific reward function $r_{\mathcal{T}}$ and $z_\pi^{\mathcal{T}}$

represents the features of the task-specific optimal policy $\pi_{\mathcal{T}}^*$ and the transition function $P_{\mathcal{T}}$. So we denote $z_\pi^{\mathcal{T}}$ as the policy embedding and $z_r^{\mathcal{T}}$ as the reward embedding of task $\mathcal{T}$. Based on Theorem 1, when the similarity metric is selected as (3), the task embedding space with the property (1) exists. Note that, the similarity metric in (3) is asymmetric, i.e., $S(z^{\mathcal{T}_1}, z^{\mathcal{T}_2}) \neq S(z^{\mathcal{T}_2}, z^{\mathcal{T}_1})$. To achieve the property in (1), we design $S(z^{\mathcal{T}_1}, z^{\mathcal{T}_2})$ in (3) to reflect the fitness of $\pi_{\mathcal{T}_2}^*$ to task $\mathcal{T}_1$. In particular, (3) employs the cosine similarity between the reward embedding $z_r^{\mathcal{T}_1}$ (represents the reward $r_{\mathcal{T}_1}$ of $\mathcal{T}_1$) and the policy embedding $z_\pi^{\mathcal{T}_2}$ (represents the optimal policy $\pi_{\mathcal{T}_2}^*$ of $\mathcal{T}_2$) to represent the fitness of $\pi_{\mathcal{T}_2}^*$ to $\mathcal{T}_1$ i.e., $J_{\mathcal{T}_1}(\pi_{\mathcal{T}_2}^*)$. Then, the equivalence in (1) can be established. As a result, the fitness of $\pi_{\mathcal{T}_2}^*$ to $\mathcal{T}_1$ may be different from that of $\pi_{\mathcal{T}_1}^*$ to $\mathcal{T}_2$, which lead to the asymmetry.

# 5. Meta-Training with Preference-Order-Preserving Task Embedding

The section introduces the meta-training with the preference-order-preserving task embedding. Section 5.1 introduces the network structure of module (i) in Figure 1 for the meta-training, which includes the preference-order-preserving encoder and the policy network conditional on the task embeddings. Section 5.2 presents the composition of the task context. Section 5.3 introduces the recursive sampling and training procedure for the meta-training.

## 5.1. Encoder-Decoder

As stated in Section 3, during the meta-training, we aim to train an encoder to map the context $c^{\mathcal{T}}$ into the task embedding $z^{\mathcal{T}}$ and train a decoder from $z^{\mathcal{T}}$ to the task-specific policy, i.e., a policy $\pi_\theta(\cdot|z^{\mathcal{T}})$ that conditions on $z^{\mathcal{T}}$. Moreover, the embedding space is expected to be preference-order-preserving, i.e., it should satisfy Property 1 shown in Section 4. In this section, we design a network to achieve this goal, which holds a structure shown in Figure 3.

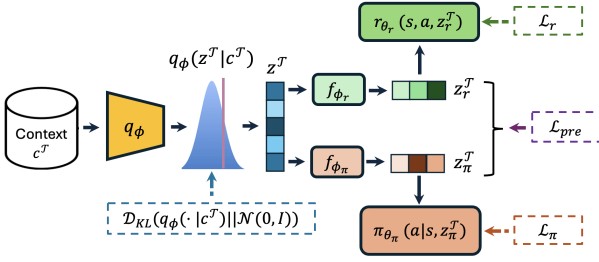

*Figure 3.* Encoder-decoder network and imposed loss functions, where using the KL divergence loss $\mathcal{D}_{KL}$ for enforcing the posterior distribution $q_\phi(\cdot|c^{\mathcal{T}})$ to $\mathcal{N}(0, I)$, the reward reconstruction loss $\mathcal{L}_r$ for recovering the true reward $r_{\mathcal{T}}$ by $r_{\theta_r}(\cdot, z^{\mathcal{T}_r})$, the policy loss $\mathcal{L}_\pi$ for recovering optimal policies by $\pi_{\theta_\pi}(\cdot, z^{\mathcal{T}_\pi})$, and the preference loss $\mathcal{L}_{pre}$ for enforcing $z^{\mathcal{T}}$ to satisfy Property 1.

We build the encoder based on the variational auto-encoder (VAE) (Kingma & Welling, 2013; Rezende et al., 2014; Alemi et al., 2016), which is primarily designed for generative models and has been widely used in context-based meta-RL methods (Rakelly et al., 2019; Zintgraf et al., 2020). Specifically, we train an inference network $q_\phi$, parameterized by $\phi$, to encode the context $c^{\mathcal{T}}$ to a distribution within the embedding space $q_\phi(\cdot|c^{\mathcal{T}})$. The task embedding $z^{\mathcal{T}}$ is sampled from the distribution $q_\phi(\cdot|c^{\mathcal{T}})$ and then is mapped to the preference-order-preserving task embedding space by two networks $f_{\phi_r}$ and $f_{\phi_\pi}$, parameterized by $\phi_r$ and $\phi_\pi$, where the reward embedding $z_r^{\mathcal{T}} = f_{\phi_r}(z^{\mathcal{T}})$ and the policy embedding $z_\pi^{\mathcal{T}} = f_{\phi_\pi}(z^{\mathcal{T}})$. The reward embedding $z_r^{\mathcal{T}}$ is used in the conditional reward function $r_{\theta_r}(\cdot|z_r^{\mathcal{T}})$ to reconstruct the reward function $r_{\mathcal{T}}$ and the policy embedding $z_\pi^{\mathcal{T}}$ is used in the conditional policy $\pi_{\theta_\pi}(\cdot|z_\pi^{\mathcal{T}})$ to recover the task-specific optimal policy $\pi_{\mathcal{T}}^*$, while $z_r^{\mathcal{T}}$ and $z_\pi^{\mathcal{T}}$ are expected to satisfy Property 1 for any task $\mathcal{T}$.

To achieve the above goal, we design the following optimization problem to train the networks in Figure 3 with parameters $\Phi = [\phi, \phi_r, \phi_\pi, \theta_r, \theta_\pi]$:

$$\min_\Phi \; \mathbb{E}_{\mathcal{T}, \mathcal{T}_1, \mathcal{T}_2 \sim \mathbb{P}(\Gamma)}[D_{\mathrm{KL}}(q_\phi(\cdot|c^{\mathcal{T}})\|\mathcal{N}(0, I)) + \beta_r \mathcal{L}_r(\Phi, \mathcal{T})$$
$$+ \beta_\pi \mathcal{L}_\pi(\Phi, \mathcal{T}) + \beta_{pre}\mathcal{L}_{pre}(\Phi, \mathcal{T}, \mathcal{T}_1, \mathcal{T}_2)].$$

The objective function includes four loss terms. (i) The KL divergence loss $D_{\mathrm{KL}}(q_\phi(\cdot|c^{\mathcal{T}}) \| \mathcal{N}(0, I))$ enables the prior distribution on the embedding space, the normal distribution $\mathcal{N}(0, I)$ to approximate the posterior distribution $q_\phi(\cdot|c^{\mathcal{T}})$, as used in VAE. (ii) The reward reconstruction loss aims to reconstruct the true reward function $r_{\mathcal{T}}$ by the reword network $r_{\theta_r}$ under the task reward embedding $z_r^{\mathcal{T}}$, which is defined as $\mathcal{L}_r(\Phi, \mathcal{T}) \triangleq$

$$\mathbb{E}_{z^{\mathcal{T}} \sim q_\phi(\cdot|c^{\mathcal{T}}),(s,a)\sim c^{\mathcal{T}}}[(r_{\mathcal{T}}(s, a) - r_{\theta_r}(s, a|z_r^{\mathcal{T}}))^2] \quad (4)$$

with $z_r^{\mathcal{T}} = f_{\phi_r}(z^{\mathcal{T}})$; (iii) The policy reconstruction loss aims to reconstruct the task-specific optimal policy $\pi_{\mathcal{T}}^*$ by the policy network $\pi_{\theta_\pi}$ under the task policy embedding $z_\pi^{\mathcal{T}}$, which is defined as

$$\mathcal{L}_\pi(\Phi, \mathcal{T}) \triangleq \mathbb{E}_{z^{\mathcal{T}} \sim q_\phi(\cdot|c^{\mathcal{T}})}[D_{KL}(\pi_{\theta_\pi}(\cdot|z_\pi^{\mathcal{T}})\|\pi_{\mathcal{T}}^*)] \quad (5)$$

with $z_\pi^{\mathcal{T}} = f_{\phi_\pi}(z^{\mathcal{T}})$; (iv) The preference loss enforces $z^{\mathcal{T}}$ to satisfy Property 1 and defined as $\mathcal{L}_{pre}(\Phi, \mathcal{T}, \mathcal{T}_1, \mathcal{T}_2) \triangleq$

$$\mathbb{E}_{z^{\mathcal{T}}, z^{\mathcal{T}_1}, z^{\mathcal{T}_2} \sim q_\phi(\cdot|c^{\mathcal{T}}), q_\phi(\cdot|c^{\mathcal{T}_1}), q_\phi(\cdot|c^{\mathcal{T}_2})}\big[D_{KL}\big(\mathbb{I}[\tau^{\mathcal{T}}(\pi_{\mathcal{T}_1}^*)$$
$$\succ_{\mathcal{T}} \tau^{\mathcal{T}}(\pi_{\mathcal{T}_2}^*)] \,\|\, \Pr[S(z^{\mathcal{T}}, z^{\mathcal{T}_1}) > S(z^{\mathcal{T}}, z^{\mathcal{T}_2})]\big)\big], \quad (6)$$

where $\tau^{\mathcal{T}}(\pi_{\mathcal{T}_1}^*)$ and $\tau^{\mathcal{T}}(\pi_{\mathcal{T}_2}^*)$ are trajectories generated by $\pi_{\mathcal{T}_1}^*$ and $\pi_{\mathcal{T}_2}^*$ on $\mathcal{T}$. Here, $\mathbb{I}[\tau^{\mathcal{T}}(\pi_{\mathcal{T}_1}^*) \succ_{\mathcal{T}} \tau^{\mathcal{T}}(\pi_{\mathcal{T}_2}^*)]$ is the ground-truth preference and $\Pr[S(z^{\mathcal{T}}, z^{\mathcal{T}_1}) > S(z^{\mathcal{T}}, z^{\mathcal{T}_2})]$ is the preference prediction based on Property 1. Following

the Bradley-Terry model (Bradley & Terry, 1952), we use the predicted probability

$$
\begin{aligned}
&\Pr[S(z^{\mathcal{T}}, z^{\mathcal{T}_1}) > S(z^{\mathcal{T}}, z^{\mathcal{T}_2})] \\
&= \frac{\exp\left(S(z^{\mathcal{T}}, z^{\mathcal{T}_1})\right)}{\exp\left(S(z^{\mathcal{T}}, z^{\mathcal{T}_1})\right) + \exp\left(S(z^{\mathcal{T}}, z^{\mathcal{T}_2})\right)}.
\end{aligned}
\tag{7}
$$

So, (6) minimizes the distance between the true preference and the preference predictor to achieve Property 1.

### 5.2. Context design

In this section, we introduce the composition of the task context $c^{\mathcal{T}}$, which is subsequently encoded to the task embedding $z^{\mathcal{T}}$. As discussed in Section 5.1 and Figure 3, the task embedding $z^{\mathcal{T}}$ is used to recover both the reward function and the optimal policy. Therefore, we expect the context $c^{\mathcal{T}}$ to include two key pieces of information: (i) the mapping from the state transition to the reward, i.e., $(s^{(t)}, a^{(t)}, s^{(t+1)}) \to r_{\mathcal{T}}^{(t)}$, (ii) the optimal policy, i.e., $s^{(t)} \to \pi_{\mathcal{T}}^*(s^{(t)})$. Accordingly, we consider the task context as

$$
c^{\mathcal{T}} = \{(s^{(t)}, \hat{a}^{(t)}, s^{(t+1)}, r_{\mathcal{T}}^{(t)}, Q_{\mathcal{T}}^{\pi_{\mathcal{T}}^*}(s^{(t)}, \hat{a}^{(t)})\}_{t=1}^{H} \tag{8}
$$

where $\hat{a}^{(t)} \sim \pi_{\mathcal{T}}^*(\cdot|s^{(t)})$ and $Q_{\mathcal{T}}^{\pi_{\mathcal{T}}^*}(s^{(t)}, \hat{a}^{(t)})$ is the optimal value on $(s^{(t)}, \hat{a}^{(t)})$. Then, the information about the optimal policy is incorporated in $\hat{a}^{(t)}$ and $Q_{\mathcal{T}}^{\pi_{\mathcal{T}}^*}$.

In existing context-based meta-RL methods, such as PEARL (Rakelly et al., 2019), the context of PEARL is defined as $c^{\mathcal{T}} = \{(s^{(t)}, a^{(t)}, s^{(t+1)}, r_{\mathcal{T}}^{(t)})\}_{t=1}^{H}$ where $a^{(t)}$ is not sampled from the optimal policy $\pi_{\mathcal{T}}^*(\cdot|s^{(t)})$. As a result, it does not include information about the optimal policy and thus performs worse than (8) (As tested in Section 7). More importantly, the task context in (8) is particularly designed for the meta-RL with adaptation from human feedback. Specifically, (8) cannot be used to meta-RL, because the optimal action $\hat{a}^{(t)}$ and the optimal value $Q_{\mathcal{T}}^{\pi_{\mathcal{T}}^*}(s^{(t)}, \hat{a}^{(t)})$ cannot be obtained for a new given task and therefore cannot be encoded to $z^{\mathcal{T}}$. In contrast, as indicated in Figure 1, after the meta-training stage of the proposed method POEM, i.e., the training of the networks in Figure 3 is done, the encoder $q_\phi$ and the context $c^{\mathcal{T}}$ will be not used during the meta-test phase. Instead, the human feedback data is used to infer the task embedding $z^{\mathcal{T}}$ to obtain the policy. Therefore, we can use the context in (8) to be as informative as possible to train a well-structured embedding space.

### 5.3. Recursive sampling and training procedure

When using the network structure and the loss functions in Section 5.1 and the context in Section 5.2 for the meta-training, an issue is that the task-specific optimal policy $\pi_{\mathcal{T}}^*$ in (5), (6) and (8) is not available. To address this issue, we employ a recursive sampling and training pro-

cedure. In specific, we first replace the policy reconstructive loss in (5) with the policy optimization loss, i.e., $\mathcal{L}_\pi(\Phi, \mathcal{T}) \triangleq \mathbb{E}_{z^{\mathcal{T}} \sim q_\phi(\cdot|c^{\mathcal{T}})}[-J_{\mathcal{T}}(\pi_{\theta_\pi}(\cdot|z_\pi^{\mathcal{T}}))]$, such that the conditional policy network $\pi_{\theta_\pi}(\cdot, z^{\mathcal{T}_\pi})$ can be gradually optimized toward the task-specific optimal policy $\pi_{\mathcal{T}}^*$. Next, as illustrated in Figure 4, at epoch $n$ of the meta-training, we replace the optimal policy $\pi_{\mathcal{T}}^*$ in the preference loss (6) and the context (8) by the policy $\pi_{\theta_\pi^{(n-1)}}(\cdot, z^{\mathcal{T}_\pi,(n-1)})$, where both the policy network $\pi_{\theta_\pi^{(n-1)}}$ and the embedding $z^{\mathcal{T}_\pi,(n-1)}$ are obtained from the last epoch $n-1$.

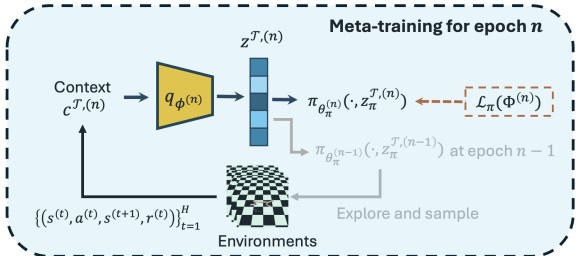

*Figure 4.* Recursive sampling and training procedure.

The recursive procedure continuously optimizes the policy, while using the current policy as the approximation for the optimal policy to train the encoder and sample the context. When the conditional policy network $\pi_{\theta_\pi}(\cdot, z_\pi^{\mathcal{T}})$ converges to the task-specific optimal policy $\pi_{\mathcal{T}}^*$, the sampled context $c^{\mathcal{T}}$ is the optimal trajectory in (8) and the employed preference loss is exactly (6).

Finally, we summarize the meta-training algorithm in this section as Algorithm 1. In Algorithm 1, we apply the recursive sampling (line 16) and training (lines 4-14) for the meta-training. Following the suggestion in context-based meta-RL methods (Rakelly et al., 2019; Fu et al., 2021; Yu et al., 2024), we employ the soft actor-critic (Haarnoja et al., 2018) for policy optimization. For the encoder $q_\phi$, we employ the permutation-invariant probabilistic network architecture used in PEARL (Rakelly et al., 2019).

## 6. Adaptation from Human Feedback by Task Embedding Inference

In this section, we introduce the adaptation from human feedback algorithm, which is applied after the meta-training described in Section 5.

After the preference-order-preserving task embedding encoder $q_\phi$, $f_{\phi_r}$, $f_{\phi_\pi}$ and the policy network $\pi_{\theta_\pi}$ that are shown in Section 5 are trained, for a given task $\mathcal{T}_{new} \in \mathbb{P}(\Gamma)$, if we can obtain its task embedding $z^{\mathcal{T}_{new}}$, the task-specific policy can be derived as $\pi_{\theta_\pi}(\cdot, z_\pi^{\mathcal{T}_{new}})$. As a result, the adaptation from human feedback is to obtain $z^{\mathcal{T}_{new}}$. More importantly, the task embedding $z^{\mathcal{T}_{new}}$ should satisfy Property

**Algorithm 1** Meta-training algorithm

**Require:** Training tasks $\{\mathcal{T}_i\}_{i=1}^T$ from $\mathbb{P}(\Gamma)$; initial network parameters $\Phi^{(0)} = \{\phi^{(0)}, \phi_r^{(0)}, \phi_\pi^{(0)}, \theta_r^{(0)}, \theta_\pi^{(0)}\}$

1: Initialize replay buffers $\mathcal{B}^i$ for each training task $\mathcal{T}_i$
2: Sample a set of data $(s, a, s', r, Q(s,a))$ by the initial policy $\pi_{\theta_\pi^{(0)}}(\cdot|z_\pi^{\mathcal{T}_i,(0)})$, where $z_\pi^{\mathcal{T}_i,(0)} \sim \mathcal{N}(0, I)$ and $Q(s, a)$ is set to 0, and add them to $\mathcal{B}^i$ for each $\mathcal{T}_i$
3: **for** epoch $n = 1, \cdots, N$ **do**
4:    **for** step $m = 1, \cdots, M$ **do**
5:       **for** each $\mathcal{T}_i$ **do**
6:          Sample context $c^{\mathcal{T}_i,(n)} \sim \mathcal{B}^i$
7:          Sample $z^{\mathcal{T}_i,(n)} \sim q_{\phi^{(n)}}(\cdot|c^{\mathcal{T}_i,(n)})$
8:          Sample RL batch $b^i \sim \mathcal{B}^i$
9:          $\mathcal{L}_{KL}(\mathcal{T}_i) = D_{KL}\left(q_{\phi^{(n)}}(\cdot|c^{\mathcal{T}_i,(n)})||\mathcal{N}(0, I)\right)$
10:         Set $\mathcal{L}_\pi(\mathcal{T}_i)$ as the policy optimization loss of SAC under the policy $\pi_{\theta_\pi^{(n)}}(\cdot|z_\pi^{\mathcal{T}_i,(n)})$ on $b^i$
11:         Set $\mathcal{L}_r(\mathcal{T}_i)$ as (4) under the embedding $z^{\mathcal{T}_i,(n)}$
12:         Sample tasks $\mathcal{T}_i', \mathcal{T}_i''$, and set $\mathcal{L}_{pre}(\mathcal{T}_i, \mathcal{T}_i', \mathcal{T}_i'')$ as (6) under the embeddings $z^{\mathcal{T}_i,(n)}$, $z^{\mathcal{T}_i',(n)}$, $z^{\mathcal{T}_i'',(n)}$
13:       **end for**
14:       Optimize $\Phi^{(n)}$ by minimizing $\sum_i \mathcal{L}_{KL}(\mathcal{T}_i) + \beta_r\mathcal{L}_r(\mathcal{T}_i) + \beta_\pi\mathcal{L}_\pi(\mathcal{T}_i) + \beta_{pre}\mathcal{L}_{pre}(\mathcal{T}_i, \mathcal{T}_i', \mathcal{T}_i'')$
15:    **end for**
16:    Clear $\mathcal{B}^i$ and sample a set of data $(s, a, s', r, Q(s,a))$ by $\pi_{\theta_\pi^{(n)}}(\cdot|z_\pi^{\mathcal{T}_i,(n)})$ and add them to $\mathcal{B}^i$ for each $\mathcal{T}_i$
17:    $\Phi^{(n+1)} \leftarrow \Phi^{(n)}$
18: **end for**

---

**Algorithm 2** Adaptation from human feedback

**Require:** Learned parameters $\Phi = \{\phi, \phi_r, \phi_\pi, \theta_r, \theta_\pi\}$, maximal preference queries $K$, error tolerance model $\epsilon$ defined in (9).

1: Preference query set $\mathcal{Q}_0 \leftarrow \emptyset$
2: **for** $k = 1, \cdots, K$ **do**
3:    The candidate embedding set $\mathcal{Z}_k \leftarrow \emptyset$
4:    **while** $|\mathcal{Z}_k| \leq N_z$ **do**
5:       Sample $z \sim \mathcal{N}(0, I)$
6:       **if** $S(z, z') >_\epsilon S(z, z'')$ for all $(z', z'') \in \mathcal{Q}_k$ **then**
7:          Add $z$ to $\mathcal{Z}_k$
8:       **end if**
9:    **end while**
10:    Select $(\hat{z}', \hat{z}'')$ from $\mathcal{Z}_k$ by (10)
11:    Query the preference oracle on the new task $\mathcal{T}_{new}$ for trajectories $\tau' = \tau^{\mathcal{T}_{new}}(\pi_{\theta_\pi}(\cdot|\hat{z}_\pi'))$ and $\tau'' = \tau^{\mathcal{T}_{new}}(\pi_{\theta_\pi}(\cdot|\hat{z}_\pi''))$
12:    **if** $\tau' \succ_{\mathcal{T}_{new}} \tau''$ **then**
13:       $\mathcal{Q}_k = \mathcal{Q}_{k-1} \cup (\hat{z}', \hat{z}'')$
14:    **else**
15:       $\mathcal{Q}_k = \mathcal{Q}_{k-1} \cup (\hat{z}'', \hat{z}')$
16:    **end if**
17:    Compute the policy $\pi_{\theta_\pi}(\cdot|z_\pi^k)$ for task $\mathcal{T}_{new}$ where $z^k = \arg\max_{z \in \mathcal{Z}_k} \sum_{(z',z'') \in \mathcal{Q}_k} \log \Pr[S(z, z') > S(z, z'')]$.
18: **end for**

---

1, i.e., for the task $\mathcal{T}_{new}$, the task-specific policy $\pi_{\theta_\pi}(\cdot, z_\pi^{\mathcal{T}})$ of $\mathcal{T}$ is preferred if the similarity between $z^{\mathcal{T}_{new}}$ and $z^{\mathcal{T}}$ is higher. Leveraging this property, we deploy two policies $\pi_{\theta_\pi}(\cdot, z_\pi^{\mathcal{T}_1})$ and $\pi_{\theta_\pi}(\cdot, z_\pi^{\mathcal{T}_2})$ on $\mathcal{T}_{new}$, query the preference oracle to obtain the preference ranking, and thereby determine whether $z^{\mathcal{T}_{new}}$ is more similar to $z^{\mathcal{T}_1}$ or $z^{\mathcal{T}_2}$, which positions $z^{\mathcal{T}_{new}}$ in a half-space. Repeat the above procedure with different policy pairs, we can progressively position the $z^{\mathcal{T}_{new}}$ more precisely in the task embedding space. The algorithm of inferring the task embedding $z^{\mathcal{T}_{new}}$ is stated in Algorithm 2.

In Algorithm 2, we continuously query the preference oracle on $\mathcal{T}_{new}$ then add the preference query set $\mathcal{Q}_k$ (lines 11-16). Simultaneously, in lines 4-9, we sample the task embedding $z$ from $\mathcal{N}(0, I)$, and only keep the task embedding candidates that align with all preference queries in the query set, i.e., $z$ with $S(z, z') > S(z, z'')$ for that the policy on $z'$ is preferred than the policy on $z''$. Note that the human preference queries during the adaptation from human feedback may include errors, and Property 1 may not be strictly satisfied for the learned preference-order-preserving encoder, especially when two query policies are similar. So, we relax

Property 1 to incorporate a preference error tolerance model $\epsilon$. Specifically, if the policy on $z'$ is preferred than the policy on $z''$, then $S(z, z') >_\epsilon S(z, z'')$, which is defined as

$$\frac{\exp(S(z, z'))}{\exp(S(z, z')) + \exp(S(z, z''))} \geq \frac{1}{2} - \epsilon. \quad (9)$$

The condition in (9) includes all $z$ with $S(z, z') > S(z, z'')$ and $z$ with that $S(z, z')$ is close to $S(z, z'')$. It can be seen as the preference probability defined in (7), $\Pr[S(z^{\mathcal{T}}, z^{\mathcal{T}_1}) > S(z^{\mathcal{T}}, z^{\mathcal{T}_2})] \geq \frac{1}{2} - \epsilon$. In line 17, we apply the maximum likelihood over all the embedding candidates to determine the output task embedding $z^k$ and policy $\pi_{\theta_\pi}(\cdot|z_\pi^k)$.

Before querying the preference oracle, we need to decide which two trajectories are queried. In line 11, to maximize the query efficiency, we select the query trajectory pairs that can eliminate the task embeddings as many as possible, i.e.,

$$(\hat{z}', \hat{z}'') = \arg\min_{z', z'' \in \mathcal{Z}_k}(\max\{|\mathcal{Z}^{(1)}|, |\mathcal{Z}^{(2)}|\}), \quad (10)$$

where $\mathcal{Z}^{(1)} = \{z \in \mathcal{Z}_k : S(z, z') >_\epsilon S(z, z'')\}$ and $\mathcal{Z}^{(2)} = \{z \in \mathcal{Z}_k : S(z, z'') >_\epsilon S(z, z')\}$. Here, $\mathcal{Z}^{(1)}$ is the set of remaining embeddings when $z'$ is preferred and $\mathcal{Z}^{(2)}$ is that when $z''$ is preferred. Considering the worse case, i.e., one of $\mathcal{Z}^{(1)}$ and $\mathcal{Z}^{(2)}$ with the larger size will be remaining, (10) first takes the set $\max\{|\mathcal{Z}^{(1)}|, |\mathcal{Z}^{(2)}|\}$ to pick the larger set in $\mathcal{Z}^{(1)}$ and $\mathcal{Z}^{(2)}$, and then minimizes its size.

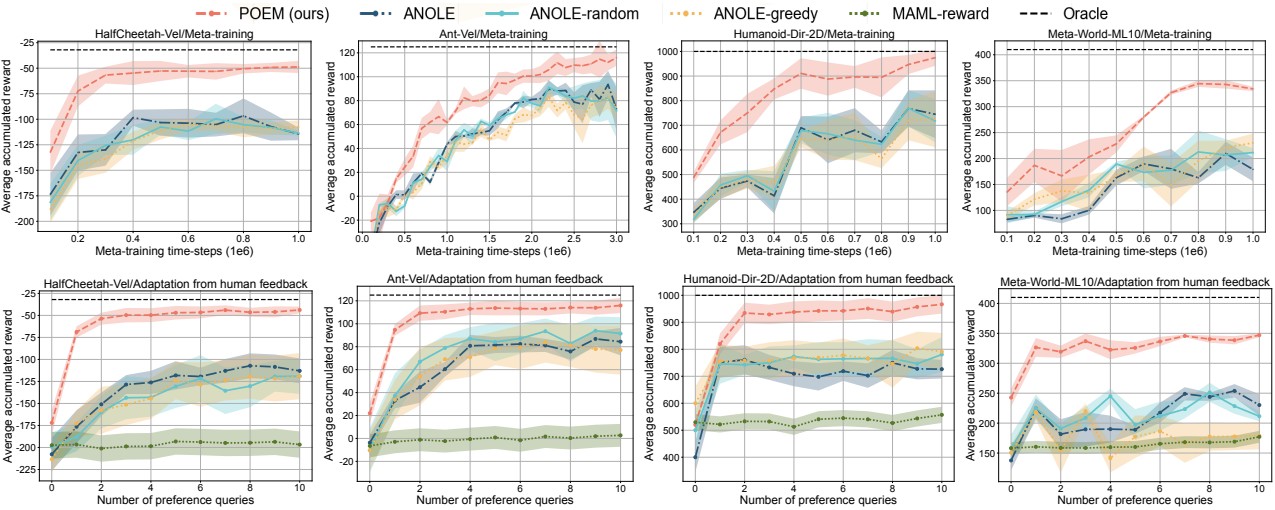

*Figure 5.* Performance on HalfCheetah-Vel, Ant-Vel, Humanoid-Dir, and MetaWorld-ML10. Average accumulated reward on test tasks v.s. samples collected during meta-training (**Top**) and during adaptation from human feedback (**Bottom**). "Oracle" denotes the performance when the reward signals are available during the meta-test.

**Theorem 2.** *Assume that (i) the task encoder $f : \Gamma \rightarrow \mathbb{R}^d$ holds that, for $\mathcal{T} \sim \mathbb{P}(\Gamma)$, $z^{\mathcal{T}} = f(\mathcal{T})$ follows the normal distribution $\mathcal{N}(0, I)$; (ii) $f_r$ and $f_\pi : \mathbb{R}^d \rightarrow \mathbb{R}^d$ are mappings such that $z_r^{\mathcal{T}} = f_r(z^{\mathcal{T}})$ and $z_\pi^{\mathcal{T}} = f_\pi(z^{\mathcal{T}})$ satisfy Property 1; (iii) the policy network with $\theta_\pi$ is optimal, i.e., $\pi_{\theta_\pi}(\cdot | z_\pi^{\mathcal{T}}) = \pi_{\mathcal{T}}^*$. Suppose the preference oracle in Algorithm 2 on task $\mathcal{T}_{new}$ holds the error at most $\epsilon$, i.e., the condition in (9) is satisfied. Let $Z_k$ be the random variable sampled as same as $\mathcal{Z}_k$ in Algorithm 2. Let $Z$ be a random variable following $\mathcal{N}(0, I)$. Under certain mild conditions on the mappings $f_r$ and $f_\pi$, we have*

   *(a) The probability density function (PDF) of $Z_k$ at $z^{\mathcal{T}_{new}}$ has $P(Z_k = z^{\mathcal{T}_{new}}) \geq \frac{P(Z=z^{\mathcal{T}_{new}})}{C_1 \cdot \left(\frac{1}{2}\right)^k + C_2 \log\left(\frac{1+2\epsilon}{1-2\epsilon}\right)}$;*

   *(b) When $\epsilon = 0$, $Z_k \xrightarrow{d} z^{\mathcal{T}_{new}}$, i.e., $Z_k$ converges to $z^{\mathcal{T}_{new}}$ in distribution.*

*Here, $C_1$ and $C_2$ are the constants, and $z^{\mathcal{T}_{new}}$ is the embedding of $\mathcal{T}_{new}$ such that $J_{\mathcal{T}_{new}}(\pi_{\theta_\pi}(\cdot | z_\pi^{\mathcal{T}_{new}})) = J_{\mathcal{T}_{new}}(\pi_{\mathcal{T}}^*)$.*

The full statements of the assumption on $f_r$ and $f_\pi$, the constants $C_1$ and $C_2$ in Theorem 2, and the proofs of Theorem 2 are provided in Appendix B.2. In Theorem 2, we derive the performance guarantee for Algorithm 2. Theorem 2 first assumes the encoder-decoder network shown in Figure 3 is well-trained. Specifically, (i) states that the posterior distribution is the normal distribution; (ii) requires that Property 1 holds, and (iii) ensures that the optimal policy is accurately reconstructed. Theorem 2 defines the candidate embedding distribution $Z_k$, from which the candidate embedding set $\mathcal{Z}_k$ in Algorithm 2 is sampled, i.e., $\mathcal{Z}_k \sim Z_k$. It is shown

that the probability density of $Z_k$ on $z^{\mathcal{T}_{new}}$ increases monotonically. If the preference oracle has no error, i.e., $\epsilon = 0$, then $P(Z_k = z^{\mathcal{T}_{new}}) = \mathcal{O}(2^k)$, which increases exponentially, and the candidate distribution $Z_k$ converges to $z^{\mathcal{T}_{new}}$ in distribution. To the best of our knowledge, Theorem 2 is the first to provide a performance guarantee for meta-RL with adaptation from human feedback.

## 7. Experiment

In the experiments, we evaluate the performance of the proposed approach, POEM, by addressing the following research questions: (i) How does POEM perform compared to baseline methods of meta-RL with adaptation from human feedback? (ii) Can POEM effectively handle errors in the preference oracle? (iii) How do individual components of PEOM impact its performance?

**Experiment settings.** We conduct experiments on nine scenarios in two benchmarks centered around robotic continuous control tasks. The first benchmark is **MuJoCo**, involving robotic locomotion tasks simulated by the MuJoCo physics engine (Todorov et al., 2012), which is a widely adopted platform in meta-RL works (Finn et al., 2017; Rothfuss et al., 2019; Rajeswaran et al., 2019) to test the performance of few-shot policy adaptation. The second is **MetaWorld** (Yu et al., 2020), a benchmark suite for robotic tabletop manipulation, which includes a diverse set of motion patterns for robotic arms and is recognized as a more challenging benchmark for meta-RL. Detailed descriptions for the two benchmarks can be found in Appendix C.1. Across the nine scenarios, each scenario consists of a task

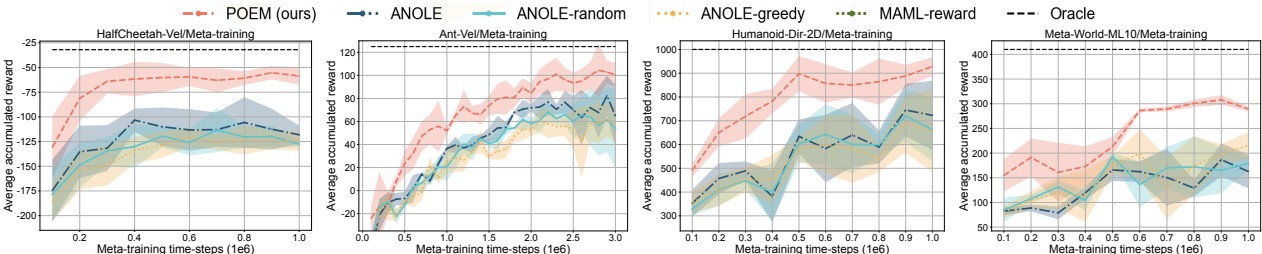

*Figure 6.* Performance evaluation under a noisy preference oracle. Average accumulated reward on test tasks v.s. samples collected during meta-training on HalfCheetah-Vel, Ant-Vel, Humanoid-Dir, and MetaWorld-ML10. "Oracle" denotes the performance when the reward signals are available during the meta-test.

family, where each task within the family is characterized by its dynamic system and reward function. The details of the task settings can be found in Appendix C.2.

**Baseline algorithms.** We compare our method, POEM, with four state-of-the-art baseline methods for meta-RL with adaptation from human feedback: (a) probabilistic context-based meta-RL with three strategies for the adaptation from human feedback: ANOLE, Greedy Binary Search, and Random Query, which are introduced in (Ren et al., 2022); and (b) MAML-reward (Joey Hejna, 2023), which applies MAML (Finn et al., 2017) to train a meta reward model to adapt the task-specific reward model. Moreover, we also compare with (c) oracle, which is obtained when the reward signals are available during the adaptation. Since MAML-reward only trains the reward model during the meta-training, there is no performance curve for the meta-training policies. To ensure a fair comparison, the data requirements for all the methods are the same during both the meta-training and the adaptation from human feedback. More details of the experimental settings are introduced in Appendix C.3.

**Performance evaluation.** Figure 5 compares the performance of POEM with the baseline methods under an accurate preference oracle on four scenarios. The results for the remaining five scenarios are shown in Appendix C.5. It is shown that the proposed method, POEM, significantly outperforms all the baseline methods, i.e. 20%-50% improvement over the best baselines in terms of the accumulated rewards during both the meta-training and the meta-test. More detailed comparisons between POEM and the baselines are shown in Appendix C.5.

**Performance under noisy preference oracle.** We evaluate POEM when the preference oracle holds noises during the adaptation from human feedback. We use the Boltzmann model, introduced in (Lee et al., 2021a), to model the human preference errors. Figure 6 illustrates the performance of POEM under a noisy preference oracle across four scenarios. The results indicate that POEM achieves performance

comparable to that observed with an accurate preference oracle, as shown in Figure 5, highlighting its robustness to preference errors. Notably, (Ren et al., 2022) has demonstrated ANOLE's strong performance in handling preference noises, and the proposed method significantly outperforms ANOLE in these scenarios. Detailed descriptions of the reward model, comparison results for more scenarios, and detailed analysis are shown in Appendix C.5.

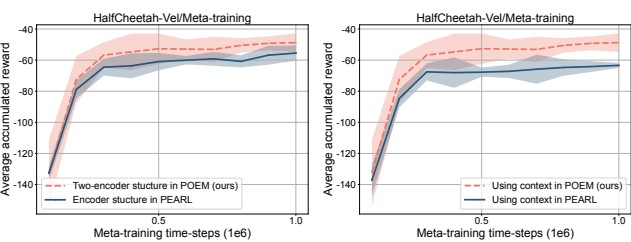

*Figure 7.* Ablation studies on Halfchehta-Vel.

**Ablation studies.** We conduct ablation studies to test the impact of components designed for POEM, including (i) the two-encoder structure, i.e., the policy/reward embedding encoders, with distance metric in (3) (Section 5.1) and (ii) the context design (Section 5.2). Detailed descriptions of the comparisons between the components designed for POEM and those used in previous works are shown in Appendix C.7. Figures 7, 12, and 13 demonstrate the effectiveness of the components (i) and (ii) designed for POEM.

## 8. Conclusion

In this paper, we propose a novel meta-RL algorithm, adaptation via preference-order-preserving embedding (POEM) for meta-RL with adaptation from human feedback. The key idea of POEM is to train a task encoder such that the task embedding space holds the preference-order-preserving property. We provide a theoretical analysis regarding the convergence of the adaptation process and empirically demonstrate its superior effectiveness.

## Acknowledgements

This work is partially supported by the National Science Foundation through grants ECCS 1846706 and ECCS 2140175. We would like to thank the reviewers for their constructive and insightful suggestions.

## Impact Statement

This paper presents work whose goal is to advance the field of Machine Learning. There are many potential societal consequences of our work, none of which we feel must be specifically highlighted here.

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

# Appendix for "Meta-Reinforcement Learning with Adaptation from Human Feedback via Preference-Order-Preserving Task Embedding"

## A. Related works

**From RLHF to meta-RL with adaptation from human feedback.** RL from human feedback (RLHF) (Wirth et al., 2017; Kaufmann et al., 2023; Casper et al., 2023) has been extensively studied to align the objective of an RL task with human intentions, while the reward function of the task is not directly accessible. A mainstream of the existing methods continuously inquires humans to obtain human feedback data, e.g., the binary trajectory comparisons or trajectory rankings, and use the data to infer the reward function, then optimize the policy based on the inferred reward function (Christiano et al., 2017; Ibarz et al., 2018; Xu et al., 2020; Chakraborty et al., 2024). Several methods have been developed to improve the efficiency of data utilization, including optimizing the query selections (Biyik & Sadigh, 2018; Biyik et al., 2020; Chakraborty et al., 2024), initializing with a better policy (Ibarz et al., 2018; Lee et al., 2021b), and improving exploration efficiency (Liang et al., 2022). Although these methods enhance data efficiency, the number of feedback queries remains substantial, often ranging from 1k to 10k (Liang et al., 2022). This paper addresses the data inefficiency issue in RLHF through meta-RL with adaptation from human feedback. Specifically, we propose pre-training a meta-model by extracting knowledge from multiple existing RL tasks. This enables the meta-model to leverage few-shot human preference data from a new task to derive a high-quality policy. By utilizing knowledge from the pre-training tasks, the number of required feedback queries for a new task can be significantly reduced to as few as 5 to 10.

**Categorization of meta-RL.** Meta-RL methods can be generally categorized into (i) optimization-based, (ii) black-box, and (iii) context-based methods (Beck et al., 2023). (i) The optimization-based meta-RL, including MAML (Finn et al., 2017) and its variants (Stadie et al., 2018; Liu et al., 2019; Xu & Zhu, 2024; 2022), has a bilevel optimization (Ji et al., 2021; Xu & Zhu, 2023) structure. In the lower-level optimization, the task-specific policy is adapted from a shared meta-policy using a policy optimization algorithm (e.g., one-step policy gradient in MAML). In the upper-level optimization, the meta-policy is optimized to maximize the performance of the adapted policy starting from the meta-policy over training tasks. (ii) Black-box meta-RL (Duan et al., 2017; Wang et al., 2016) learn an end-to-end neural network model. The model has fixed parameters during the policy adaptation, and takes the history samples of the task as the input and the task-specific policy as the output. (iii) Context-based meta-RL (Rakelly et al., 2019; Raileanu et al., 2020; Zintgraf et al., 2020) encode the samples of the tasks to the task embedding vector and decode the embedding vector to the task-specific policy. In meta-RL with adaptation from human feedback, the reward is non-accessible. However, it is indispensable in policy optimization algorithms and also provides important information within the history samples of a task. Therefore, the optimization-based and black-box methods cannot be directly applied in this setting. In contrast, in context-based meta-RL, although the encoding process is disrupted due to missing reward signals, the task embedding vector can be inferred from human preference data. Both ANOLE (Ren et al., 2022) and this paper belong to this category. However, ANOLE borrows the task embedding encoder from an existing context-based meta-RL model, and the task embedding space is not designed for preference data. In this paper, we derive a preference-ordering-preserving embedding space mapped from the task space, which is more suitable for task embedding inference from human preference data.

**Comparisons of methods for meta-RL with adaptation from human feedback.** Paper (Joey Hejna, 2023) applies a supervised meta-learning approach, MAML (Finn et al., 2017), to train a meta-reward model, and adapt it to the task-specific reward model with few-shot human preference data during the meta-test. However, the method still needs to solve the RL problem to obtain the task-specific policy under the learned task-specific reward model, which requires a large amount of data from environmental exploration. To address scenarios of the meta-RL where both human feedback data and environment exploration data are few-shot, ANOLE (Ren et al., 2022) infers the embedding vector of the task from the human preference data and derives the task-specific policy from a well-trained context-based meta-RL model. In particular, ANOLE first employs a context-based meta-RL model trained by PEARL (Rakelly et al., 2019), which provides an encoder that maps the task space to a task embedding vector space and a decoder that maps the embedding vectors to the task-specific policies. Next, ANOLE trains a conditional reward model that takes the task embedding vector as the input and the task-specific reward as the output. During the meta-test, ANOLE matches the human preference queries and the conditional reward model to infer the task embedding vector, and then decodes the vector to obtain the task-specific policy. An issue of ANOLE is that the meta-training and meta-test data for the task embedding decoder are inconsistent. Specifically, During the meta-training of PEARL, task-specific samples with reward signals are used to encode tasks to embedding vectors. However, human preference data is used to infer the embeddings during the meta-test, but is not involved during the meta-training. As a result, the task embedding space is not specifically designed for preference data, which may lead to sub-optimal performance. In

contrast, the proposed method, POEM, trains a preference-ordering-preserving task embedding encoder, which establishes a connection between the task embeddings and preference data. During the meta-test, the few-shot preference comparisons on the new task can be directly used to obtain the task embedding.

## B. Analysis and Proof

### B.1. Selection of similarity metric

B.1.1. INEFFECTIVENESS OF THE SIMPLE SIMILARITY METRIC

As claimed in Section 4, the similarity metric $S(z^{\mathcal{T}_1}, z^{\mathcal{T}_2}) \triangleq \langle z^{\mathcal{T}_1}, z^{\mathcal{T}_2} \rangle$ is not valid for the preference-ordering-preserving embedding space. The formal statement is shown in Proposition 1.

**Proposition 1.** *Consider $S(z^{\mathcal{T}_1}, z^{\mathcal{T}_2}) \triangleq \langle z^{\mathcal{T}_1}, z^{\mathcal{T}_2} \rangle$. There exists a task space $\Gamma$ such that, for any task encoder mapping $f$, there exist tasks $\mathcal{T}_0$, $\mathcal{T}_1$, and $\mathcal{T}_2 \in \Gamma$ along their embeddings $z^{\mathcal{T}_0} = f(\mathcal{T}_0)$, $z^{\mathcal{T}_1} = f(\mathcal{T}_1)$, and $z^{\mathcal{T}_2} = f(\mathcal{T}_2)$ with that both* (1) *and Property 1 are violated, i.e. the following property is violated:*

$$S(z^{\mathcal{T}_0}, z^{\mathcal{T}_1}) \geq S(z^{\mathcal{T}_0}, z^{\mathcal{T}_2}) \iff \tau^{\mathcal{T}_0}(\pi^*_{\mathcal{T}_1}) \succ_{\mathcal{T}_0} \tau^{\mathcal{T}_0}(\pi^*_{\mathcal{T}_2}).$$

*Proof.* Assume that given any task space $\Gamma$ such that, for any task encoder mapping $f$, for any tasks $\mathcal{T}_0$, $\mathcal{T}_1$, and $\mathcal{T}_2 \in \Gamma$ along their embeddings $z^{\mathcal{T}_0} = f(\mathcal{T}_0)$, $z^{\mathcal{T}_1} = f(\mathcal{T}_1)$, and $z^{\mathcal{T}_2} = f(\mathcal{T}_2)$ where the following property is satisfied:

$$\langle f(\mathcal{T}_0), f(\mathcal{T}_1) \rangle \geq \langle f(\mathcal{T}_0), f(\mathcal{T}_2) \rangle \iff \tau^{\mathcal{T}_0}(\pi^*_{\mathcal{T}_1}) \succ_{\mathcal{T}_0} \tau^{\mathcal{T}_0}(\pi^*_{\mathcal{T}_2}).$$

Consider the task distribution $\Gamma$ that, for any tasks $\mathcal{T}_0$, $\mathcal{T}_1$, and $\mathcal{T}_2 \in \Gamma$ with that (i) the initial states of $\mathcal{T}_0$, $\mathcal{T}_1$, and $\mathcal{T}_2$ is fixed, (ii) the optimal policy $\pi^*_{\mathcal{T}_0}, \pi^*_{\mathcal{T}_1}, \pi^*_{\mathcal{T}_2}$ for $\mathcal{T}_0$, $\mathcal{T}_1$, and $\mathcal{T}_2 \in \Gamma$ are deterministic. Then, we have

$$\langle f(\mathcal{T}_0), f(\mathcal{T}_1) \rangle \geq \langle f(\mathcal{T}_0), f(\mathcal{T}_2) \rangle \iff J_{\mathcal{T}_0}(\pi^*_{\mathcal{T}_1}) \geq J_{\mathcal{T}_0}(\pi^*_{\mathcal{T}_2}) \text{ for any } \mathcal{T}_0, \mathcal{T}_1, \text{ and } \mathcal{T}_2 \in \Gamma.$$

Also, we have

$$\langle f(\mathcal{T}_0), f(\mathcal{T}_1) \rangle \leq \langle f(\mathcal{T}_0), f(\mathcal{T}_2) \rangle \iff J_{\mathcal{T}_0}(\pi^*_{\mathcal{T}_1}) \leq J_{\mathcal{T}_0}(\pi^*_{\mathcal{T}_2}) \text{ for any } \mathcal{T}_0, \mathcal{T}_1, \text{ and } \mathcal{T}_2 \in \Gamma.$$

They imply that

$$\langle f(\mathcal{T}_0), f(\mathcal{T}_1) \rangle = \langle f(\mathcal{T}_0), f(\mathcal{T}_2) \rangle \iff J_{\mathcal{T}_0}(\pi^*_{\mathcal{T}_1}) = J_{\mathcal{T}_0}(\pi^*_{\mathcal{T}_2}) \text{ for any } \mathcal{T}_0, \mathcal{T}_1, \text{ and } \mathcal{T}_2 \in \Gamma.$$

If tasks $\mathcal{T}_1'$ and $\mathcal{T}_2'$ have that the reward function $r_{\mathcal{T}_1'} \neq r_{\mathcal{T}_2'}$ but $\pi^*_{\mathcal{T}_1'} = \pi^*_{\mathcal{T}_2'}$. Then, we have

$$J_{\mathcal{T}_0}(\pi^*_{\mathcal{T}_1'}) = J_{\mathcal{T}_0}(\pi^*_{\mathcal{T}_2'}) \text{ for any } \mathcal{T}_0 \in \Gamma.$$

Then,
$$\langle f(\mathcal{T}_0), f(\mathcal{T}_1') \rangle = \langle f(\mathcal{T}_0), f(\mathcal{T}_2') \rangle \text{ for any } \mathcal{T}_0 \in \Gamma.$$

Consider task $\mathcal{T}_0 = \mathcal{T}_1'$, then $||f(\mathcal{T}_1')||_2^2 = \langle f(\mathcal{T}_1'), f(\mathcal{T}_2') \rangle$, then we have $f(\mathcal{T}_1') = f(\mathcal{T}_2')$.

Next, consider the optimal policies of tasks $\mathcal{T}_0$ and $\mathcal{T}_0'$ are evaluated under tasks $\mathcal{T}_1'$ and $\mathcal{T}_2'$, we have

$$\langle f(\mathcal{T}_0'), f(\mathcal{T}_1') \rangle \geq \langle f(\mathcal{T}_0), f(\mathcal{T}_1') \rangle \iff J_{\mathcal{T}_1'}(\pi^*_{\mathcal{T}_0'}) \geq J_{\mathcal{T}_1'}(\pi^*_{\mathcal{T}_0}) \text{ for any } \mathcal{T}_0, \mathcal{T}_0' \in \Gamma.$$

$$\langle f(\mathcal{T}_0'), f(\mathcal{T}_2') \rangle \geq \langle f(\mathcal{T}_0), f(\mathcal{T}_2') \rangle \iff J_{\mathcal{T}_2'}(\pi^*_{\mathcal{T}_0'}) \geq J_{\mathcal{T}_2'}(\pi^*_{\mathcal{T}_0}) \text{ for any } \mathcal{T}_0, \mathcal{T}_0' \in \Gamma.$$

Since $f(\mathcal{T}_1') = f(\mathcal{T}_2')$, we have

$$\langle f(\mathcal{T}_0'), f(\mathcal{T}_1') \rangle \geq \langle f(\mathcal{T}_0), f(\mathcal{T}_1') \rangle \iff \langle f(\mathcal{T}_0'), f(\mathcal{T}_2') \rangle \geq \langle f(\mathcal{T}_0), f(\mathcal{T}_2') \rangle.$$

Then, we have

$$J_{\mathcal{T}_1'}(\pi^*_{\mathcal{T}_0'}) \geq J_{\mathcal{T}_1'}(\pi^*_{\mathcal{T}_0}) \iff J_{\mathcal{T}_2'}(\pi^*_{\mathcal{T}_0'}) \geq J_{\mathcal{T}_2'}(\pi^*_{\mathcal{T}_0}) \text{ for any } \mathcal{T}_0, \mathcal{T}_0' \in \Gamma.$$

However, when the reward function $r_{\mathcal{T}_1'} \neq r_{\mathcal{T}_2'}$, we can easily to find tasks $\mathcal{T}_0$ and $\mathcal{T}_0'$ such that the above statement is violated.

$\square$

B.1.2. EFFECTIVENESS OF THE PROPOSED SIMILARITY METRIC

In this subsection, we prove Theorem 1.

*Proof.* Consider the mapping $f_r$ with that

$$f_r(\mathcal{T}) = [r_{\mathcal{T}}(s_1, a_1), r_{\mathcal{T}}(s_1, a_2), \cdots, r_{\mathcal{T}}(s_1, a_{|\mathcal{A}|}), r_{\mathcal{T}}(s_2, a_1), \cdots, r_{\mathcal{T}}(s_{|\mathcal{S}|}, a_{|\mathcal{A}|})],$$

and the mapping $f_\pi$ with that

$$f_\pi(\mathcal{T}) = [\nu_{\mathcal{T}}^{\pi_{\mathcal{T}}^*}(s_1)\pi_{\mathcal{T}}^*(a_1|s_1), \nu_{\mathcal{T}}^{\pi_{\mathcal{T}}^*}(s_1)\pi_{\mathcal{T}}^*(a_2|s_1), \cdots, \nu_{\mathcal{T}}^{\pi_{\mathcal{T}}^*}(s_1)\pi_{\mathcal{T}}^*(a_{|\mathcal{A}|}|s_1)), \cdots, \nu_{\mathcal{T}}^{\pi_{\mathcal{T}}^*}(s_{|\mathcal{S}|})\pi_{\mathcal{T}}^*(a_{|\mathcal{A}|}|s_{|\mathcal{S}|}))].$$

Then, we have that, for any tasks $\mathcal{T}_0$ and $\mathcal{T}_1$, we have

$$\langle f_r(\mathcal{T}_0), f_\pi(\mathcal{T}_1) \rangle = J_{\mathcal{T}_0}(\pi_{\mathcal{T}_1}^*).$$

Then we have

$$\langle z_r^{\mathcal{T}_0}, z_\pi^{\mathcal{T}_1} \rangle \geq \langle z_r^{\mathcal{T}_0}, z_\pi^{\mathcal{T}_2} \rangle \iff J_{\mathcal{T}_0}(\pi_{\mathcal{T}_1}^*) \geq J_{\mathcal{T}_0}(\pi_{\mathcal{T}_2}^*),$$

where $z_r^{\mathcal{T}_i} = f_r(\mathcal{T}_i)$ and $z_\pi^{\mathcal{T}_i} = f_\pi(\mathcal{T}_i)$ for $i = 0, 1$ and 2. $\qquad\square$

## B.2. Performance guarantee

We first show and justify the complete assumptions for Theorem 2.

**Assumption 1.** *Assume that the following networks are applied in Algorithm 2.*

(i) *The task encoder $f : \Gamma \to \mathbb{R}^d$ holds that, for $\mathcal{T} \sim \mathbb{P}(\Gamma)$, $z^{\mathcal{T}} = f(\mathcal{T})$ follows the normal distribution $\mathcal{N}(0, I)$.*

(ii) *The following properties hold for the mappings $f_r$ and $f_\pi : \mathbb{R}^d \to \mathbb{R}^d$:*

    (ii.a) *When $z_r^{\mathcal{T}} = f_r(z^{\mathcal{T}})$ and $z_\pi^{\mathcal{T}} = f_\pi(z^{\mathcal{T}})$, Property 1 hold for $\Gamma$.*

    (ii.b) *The mapping $f_r$ is a bijection from $\mathbb{R}^d$ to $\mathbb{R}^d$, and $f_r^{-1}$ is $L_r$-Lipschitz continuous.*

    (ii.c) *The mapping $f_\pi$ is $L_\pi$-Lipschitz continuous.*

(iii) *the policy network with $\theta_\pi$ is optimal, i.e., $\pi_{\theta_\pi}(\cdot|z_\pi^{\mathcal{T}}) = \pi_{\mathcal{T}}^*$.*

**Assumption 2.** *Assume the preference oracle on task $\mathcal{T}_{new}$ used in Algorithm 2 holds the error at most $\epsilon$, i.e., the condition in (9) is satisfied.*

In Assumption 1, we assume the encoder-decoder network shown in Figure 3 is well-trained. Specifically, assumption (i) states that the posterior distribution is a normal distribution, which is enforced by the KL divergence loss $\mathcal{D}_{KL}$; assumption (ii) requires that Property 1 holds, which is encouraged by the preference loss $\mathcal{L}_{pre}$; and assumption (iii) ensures that the optimal policy is accurately reconstructed, as enforced by the policy loss $\mathcal{L}_\pi$. Moreover, we require the continuity property for the mappings $f_r$ and $f_\pi$. Note that, for different tasks, the reward functions are different, and vice versa. So, a well-trained mapping $f_r$ is a bijection, and then $f_r^{-1}$ exists. In Assumption 1, we assume the maximum error of the preference oracle on task $\mathcal{T}_{new}$ is $\epsilon$.

Next, we provide the complete statement of Theorem 2, which is shown in Theorem 3.

**Theorem 3.** *Suppose Assumptions 1 and 2 hold for Algorithm 2. Let $Z_k$ be the random variable sampled as same as $\mathcal{Z}_k$ in Algorithm 2. Let $Z$ be a random variable following $\mathcal{N}(0, I)$. Under mild conditions on $f_r$ and $f_\pi$, we have*

(a) *The probability density function (PDF) of $Z_k$ at $z^{\mathcal{T}_{new}}$ has*

$$P(Z_k = z^{\mathcal{T}_{new}}) \geq \frac{P(Z = z^{\mathcal{T}_{new}})}{\left(1 - \frac{\sqrt{2}L_r}{\sqrt{\pi}} \log\left(\frac{1+2\epsilon}{1-2\epsilon}\right)\right) \cdot \left(\frac{1}{2}\right)^k + \frac{\sqrt{2}L_r}{\sqrt{\pi}} \log\left(\frac{1+2\epsilon}{1-2\epsilon}\right)}.$$

(b) *When $\epsilon = 0$, $Z_k$ converges to $z^{\mathcal{T}_{new}}$ in distribution, i.e., $Z_k \xrightarrow{d} z^{\mathcal{T}_{new}}$.*

*Here, $z^{\mathcal{T}_{new}}$ is the embedding of $\mathcal{T}_{new}$ such that $J_{\mathcal{T}_{new}}(\pi_{\theta_\pi}(\cdot|z_\pi^{\mathcal{T}_{new}})) = J_{\mathcal{T}_{new}}(\pi_{\mathcal{T}}^*)$.*

*Proof of (a).* Consider the space of $z$ define by:

$$\frac{\exp\left(S(z, z')\right)}{\exp\left(S(z, z')\right) + \exp\left(S(z, z'')\right)} \geq \frac{1}{2} - \epsilon.$$

the space of $z$ can be simplified as

$$S(z, z') - S(z, z'') \geq \log\left(\frac{1 + 2\epsilon}{1 - 2\epsilon}\right).$$

Consider the distance metric is $S(z', z'') = \langle z'_r, z''_\pi \rangle$, we can derive the space for the task reward embedding $z_r$ as a half-space specified by the following linear inequality:

$$\langle z_r, z''_\pi - z'_\pi \rangle \geq \log\left(\frac{1 + 2\epsilon}{1 - 2\epsilon}\right). \tag{11}$$

We denote the half-space specified in (11) with $z'_\pi$, $z''_\pi$, and $\epsilon$ as $H(z'_\pi, z''_\pi, \epsilon)$.

Let $Z_r = f_r(Z)$ where $Z$ is a random variable following $\mathcal{N}(0, I)$. Define the function $g(z'_\pi, z''_\pi) = \Pr[Z_r \in H(z'_\pi, z''_\pi, 0)]$. Since $f_r^{-1}$ is Lipschitz continuous and the probability density of $Z$ on any point is finite, then the probability density of $Z_r$ on any point is finite, then $g(z'_\pi, z''_\pi)$ is continuous.

Next, we have $g(z'_\pi, z''_\pi) = 1 - g(z''_\pi, z'_\pi)$, $0 \leq g(z'_\pi, z''_\pi) \leq 1$, and $0 \leq g(z''_\pi, z'_\pi) \leq 1$. Without loss of generality, we assume that $g(z'_\pi, z''_\pi) \geq \frac{1}{2}$ and $g(z''_\pi, z'_\pi) \leq \frac{1}{2}$. From the mean value theorem, since $g(z'_\pi, z''_\pi)$ is continuous, there exists a pair $(\bar{z}'_\pi, \bar{z}''_\pi)$ such that $g(\bar{z}'_\pi, \bar{z}''_\pi) = \frac{1}{2}$ and $g(\bar{z}''_\pi, \bar{z}'_\pi) = \frac{1}{2}$. We choose $(\bar{z}'_\pi, \bar{z}''_\pi)$ to generate the half-space $H(\bar{z}'_\pi, \bar{z}''_\pi, \epsilon)$, and aim to derive the upper bound for $\Pr[Z_r \in H(\bar{z}'_\pi, \bar{z}''_\pi, \epsilon)]$ and $\Pr[Z_r \in H(\bar{z}''_\pi, \bar{z}'_\pi, \epsilon)]$.

Since the mapping $f_r^{-1}$ is $L_r$-Lipschitz continuous and $Z$ is a random variable following $\mathcal{N}(0, I)$, we have

$$\Pr[Z_r \in H(\bar{z}'_\pi, \bar{z}''_\pi, \epsilon)/H(\bar{z}'_\pi, \bar{z}''_\pi, 0)] \leq \frac{L_r}{\sqrt{2\pi}} \log\left(\frac{1 + 2\epsilon}{1 - 2\epsilon}\right).$$

Then,

$$\Pr[Z_r \in H(\bar{z}'_\pi, \bar{z}''_\pi, \epsilon)] \leq \frac{1}{2} + \frac{L_r}{\sqrt{2\pi}} \log\left(\frac{1 + 2\epsilon}{1 - 2\epsilon}\right).$$

Similarly, we have

$$\Pr[Z_r \in H(\bar{z}''_\pi, \bar{z}'_\pi, \epsilon)] \leq \frac{1}{2} + \frac{L_r}{\sqrt{2\pi}} \log\left(\frac{1 + 2\epsilon}{1 - 2\epsilon}\right).$$

Therefore,

$$\max\{\Pr[Z_r \in H(\bar{z}'_\pi, \bar{z}''_\pi, \epsilon)], \Pr[Z_r \in H(\bar{z}''_\pi, \bar{z}'_\pi, \epsilon)]\} \leq \frac{1}{2} + \frac{L_r}{\sqrt{2\pi}} \log\left(\frac{1 + 2\epsilon}{1 - 2\epsilon}\right).$$

In Algorithm 2, the pair $(\hat{z}', \hat{z}'')$ is generated by the following optimization problem,

$$(\hat{z}', \hat{z}'') = \arg\min_{z', z'' \in \mathcal{Z}_k} (\max\{|\mathcal{Z}^{(1)}|, |\mathcal{Z}^{(2)}|\}),$$

where $\mathcal{Z}^{(1)} = \{z \in \mathcal{Z}_k : S(z, z') >_\epsilon S(z, z'')\}$ and $\mathcal{Z}^{(2)} = \{z \in \mathcal{Z}_k : S(z, z'') >_\epsilon S(z, z')\}$. This means that

$$(\hat{z}', \hat{z}'') = \arg\min_{z', z'' \in \mathcal{Z}_k} (\max\{\Pr[Z_r \in H(z'_\pi, z''_\pi, \epsilon)], \Pr[Z_r \in H(z''_\pi, z'_\pi, \epsilon)]\}).$$

Therefore, we have

$$\max\{\Pr[Z_r \in H(\hat{z}'_\pi, \hat{z}''_\pi, \epsilon)], \Pr[Z_r \in H(\hat{z}''_\pi, \hat{z}'_\pi, \epsilon)]\} \leq \frac{1}{2} + \frac{L_r}{\sqrt{2\pi}} \log\left(\frac{1 + 2\epsilon}{1 - 2\epsilon}\right).$$

Therefore, after we generate the pair $(\hat{z}', \hat{z}'')$ for one time to sample $\mathcal{Z}_1$, there is at least probability of $\frac{1}{2} - \frac{L_r}{\sqrt{2\pi}} \log\left(\frac{1+2\epsilon}{1-2\epsilon}\right)$ is eliminated. From Assumption 2, the $z_r^{\mathcal{T}_{new}}$ will be included in $H(\hat{z}'_\pi, \hat{z}''_\pi, \epsilon)$ when $\hat{z}'$ is preferred and included in $H(\hat{z}''_\pi, \hat{z}'_\pi, \epsilon)$ when $\hat{z}''$ is preferred. Therefore, $z_r^{\mathcal{T}_{new}}$ will not be eliminated.

Denote the left probability as $P_k$, then $P_0 = 1$ and $P_1 \leq \frac{1}{2} + \frac{L_r}{\sqrt{2\pi}} \log\left(\frac{1+2\epsilon}{1-2\epsilon}\right)$. Repeat the above procedure to sample $\mathcal{Z}_2$, $\mathcal{Z}_3, \cdots, \mathcal{Z}_k$, we have

$$P_k \leq \frac{1}{2}P_{k-1} + \frac{L_r}{\sqrt{2\pi}} \log\left(\frac{1+2\epsilon}{1-2\epsilon}\right).$$

Then, we have

$$P_k \leq \left(1 - \frac{\sqrt{2}L_r}{\sqrt{\pi}} \log\left(\frac{1+2\epsilon}{1-2\epsilon}\right)\right) \cdot \left(\frac{1}{2}\right)^k + \frac{\sqrt{2}L_r}{\sqrt{\pi}} \log\left(\frac{1+2\epsilon}{1-2\epsilon}\right)$$

Then,

$$P(Z_k = z^{\mathcal{T}_{new}}) \geq \frac{P(Z = z^{\mathcal{T}_{new}})}{\left(1 - \frac{\sqrt{2}L_r}{\sqrt{\pi}} \log\left(\frac{1+2\epsilon}{1-2\epsilon}\right)\right) \cdot \left(\frac{1}{2}\right)^k + \frac{\sqrt{2}L_r}{\sqrt{\pi}} \log\left(\frac{1+2\epsilon}{1-2\epsilon}\right)}.$$

$\square$

*Proof of (b).* Consider $\epsilon = 0$. Consider any embedding $z^\delta$ such that $z^\delta \neq z^{\mathcal{T}_{new}}$, i.e., $J_{\mathcal{T}_{new}}(\pi_{\theta_\pi}(\cdot|z_\pi^\delta)) < J_{\mathcal{T}_{new}}(\pi_{\mathcal{T}}^*)$. Assume that the angle between the vectors $z_r^{\mathcal{T}_{new}}$ and $z_r^\delta$ is $\delta$, then $\delta > 0$.

Consider that the pair $(\hat{z}', \hat{z}'')$ generated in Algorithm 2, the probability of generating $(\hat{z}', \hat{z}'')$ such that

$$\langle z_r^{\mathcal{T}_{new}}, z_\pi'' - z_\pi' \rangle \geq 0 \text{ and } \langle z_r^\delta, z_\pi'' - z_\pi' \rangle \leq 0$$

is non-zero, until $z^\delta$ is eliminated from the candidate space. Therefore, we have that, if $z^\delta \neq z^{\mathcal{T}_{new}}$, then

$$\lim_{k \to \infty} P\left(Z_k = z^\delta\right) = 0.$$

Therefore, for any closed set $S$, if $z_r^{\mathcal{T}_{new}} \in S$, $\lim_{k \to \infty} P\left(Z_k \in S\right) = 1$; else $\lim_{k \to \infty} P\left(Z_k \in S\right) = 0$. Then, $Z_k$ converges to $z^{\mathcal{T}_{new}}$ in distribution, i.e., $Z_k \xrightarrow{d} z^{\mathcal{T}_{new}}$.

$\square$

## C. Experimental Supplements

All experiments are executed on a computer with a 5.20 GHz Intel Core i12 CPU and an NVIDIA RTX 4090 GPU.

### C.1. Benchmarks

**MuJoCo.** The first benchmark is the robotic locomotion benchmarks simulated using the MuJoCo simulator (Todorov et al., 2012). The benchmarks are created by (Finn et al., 2017; Rothfuss et al., 2019) and are a widely adopted platform in meta-RL (Rajeswaran et al., 2019; Raileanu et al., 2020; Xu & Zhu, 2024; Liu & Zhu, 2023b; Liu & Zhu; 2023a) and meta-RL with adaptation from human feedback (Joey Hejna, 2023; Ren et al., 2022) to test the performance of few-shot policy adaptation. Visualizations of the benchmark is shown at the top of Figure 8.

**MetaWorld.** The second benchmark is MetaWorld (Yu et al., 2020), a benchmark suite for robotic tabletop manipulation. MetaWorld includes a diverse set of motion patterns and interactions with different objects for robotic arms and is recognized as a more challenging benchmark for the few-shot policy adaptation in meta-RL. Visualizations of the benchmark is shown at the bottom of Figure 8.

### C.2. Task settings

We conduct experiments on totally nine scenarios in two benchmarks centered around robotic continuous control tasks.

We consider seven scenarios on the benchmark of MuJoCo. The dynamic systems of the agents in the seven scenarios include three different robots: HalfCheetah, Ant, and Humanoid. Visualizations of these agents in their respective environments are provided at the top of Figure 8, where Half-Cheetah (Figure 8.a) has a 17-dimensional state space and a 6-dimensional action space; Humanoid (Figure 8.b) has a 376-dimensional observation space and a 17-dimensional action space; Ant (Figure 8.c) has a 27-dimensional observation space and an 8-dimensional action space. The task settings include forward/backward

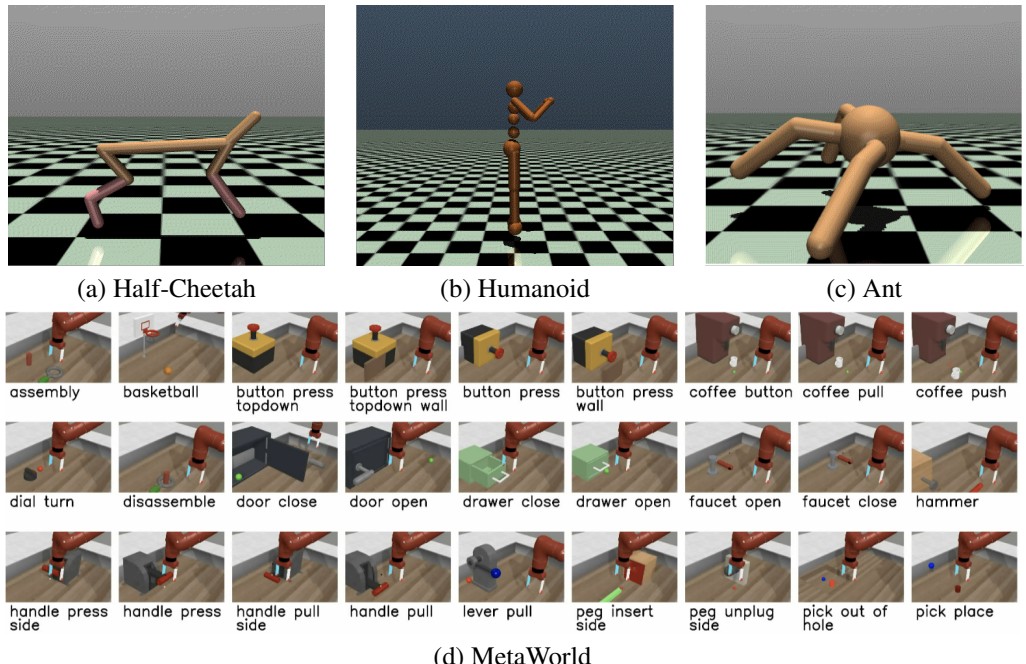

*Figure 8.* Visualization of robotic locomotion environments, including Half-Cheetah, Humanoid, and Ant, simulated by Mujoco and the MetaWorld environments.

motion (-Fwd-Back), various target velocities (-Vel), various target directions (-Dir-2D), and various target positions (-Goal). We show the detailed setting of the seven scenarios as follows.

**Half-Cheetah-Vel.** In the experiment of Half-Cheetah-Vel, the agent is Half-Cheetah, and the reward is the negative absolute value between the agent's current velocity and a goal velocity, where the goal velocity characterizes the task. The task distribution is defined by the distribution of the goal velocity, which is a uniform distribution from $0.0$ to $3.0$.

**Half-Cheetah-Fwd-Back.** In the experiment of Half-Cheetah-Fwd-Back, the agent is Half-Cheetah, and the task family includes two tasks. The reward in the first task is the agent's current velocity. The reward in the second task is the negative of the agent's current velocity.

**Ant-Vel.** In the experiment of Ant-Vel, the agent is Ant, and the reward is the negative absolute value between the agent's current velocity and a goal velocity, where the goal velocity characterizes the task. The task distribution is defined by the distribution of the goal velocity, which is a uniform distribution from $0.0$ to $3.0$.

**Ant-Fwd-Back.** In the experiment of Ant-Fwd-Back, the agent is Ant, and the task family includes two tasks. The reward in the first task is the agent's current velocity. The reward in the second task is the negative of the agent's current velocity.

**Ant-Dir-2D.** In the experiment of Ant-Dir-2D, the agent is Ant, and the reward is set as $v_y \sin\theta + v_x \cos\theta$, where $v_x$ and $v_y$ are the velocities along the $x$-axis and $y$-axis, and $\theta$ is the walking direction of the humanoid. So the reward is the velocity along the direction $\theta$. The task is characterized by the walking direction $\theta$, which is sampled uniformly from $-\pi/2$ to $\pi/2$.

**Ant-Goal.** In the experiment of Ant-Goal, the agent is Ant, and the reward is set as the negative absolute value between the agent's current location and a goal, where the goal characterizes the task. The task distribution is defined by the distribution of the goal location, which is a uniform distribution from a circle with a radius of $3.0$.

**Humanoid-Dir-2D.** In the experiment of Ant-Dir-2D, the agent is Humanoid, and the reward is set as $v_y \sin\theta + v_x \cos\theta$, where $v_x$ and $v_y$ are the velocities along the $x$-axis and $y$-axis, and $\theta$ is the walking direction of the humanoid. So the reward is the velocity along the direction $\theta$. The task is characterized by the walking direction $\theta$, which is sampled uniformly from $-\pi/2$ to $\pi/2$.

Two scenarios are conducted on MetaWorld. The scenarios in MetaWorld include task families that hold different

manipulation types (ML10) and varying operation positions (ML1). We show the detailed setting of the seven scenarios as follows.

**MetaWorld-ML1.** In the experiment of MetaWorld-ML1, the tasks family has a single type of manipulation task with varying target operation positions.

**MetaWorld-ML10.** In the experiment of MetaWorld-ML10, the task is characterized by the type of manipulation, which is shown in (d) of 8. The tasks family has 10 types of manipulation tasks where each type of manipulation task has varying target operation positions. Note that the state transition functions in different types of manipulation tasks are different.

### C.3. Experimental setting

We sample 120 training tasks for the meta-training of scenarios including Half-Cheetah-Vel, Ant-Vel, Ant-Dir-2D, Ant-Goal, Humanoid-Dir-2D, MetaWorld-ML1, MetaWorld-ML10; and sample 2 tasks for the meta-training of Half-Cheetah-Fwd-Back and Ant-Fwd-Back. We simulate the preference oracle by comparing the ground-truth trajectory return from the environments. The adaptation cannot observe the environmental rewards during meta-test and can only query the preference oracle. At the meta-test time, the meta-policy is adapted by at most 10 preference queries for each task and at most 20 trajectories without reward signals (two trajectories for a single preference query). The performance of the adapted policy is tested on 30 tasks and we evaluated 10 episodes in each task. To ensure a fair comparison, we keep all the above settings the same for all the methods during both the meta-training and the adaptation from human feedback.

### C.4. Hyper-parameters

We summarize major hyper-parameters in the following table. We use this set of hyper-parameters for all POEM's experiments. The hyper-parameters in Table 2 can be divided into two parts. The first part includes the hyper-parameters that follow the existing meta-RL (Rakelly et al., 2019) and RL (Haarnoja et al., 2018) algorithms. So we directly use the hyper-parameters in these works in our proposed method. The second part is new and specific to the proposed method, including the coefficients of the three loss terms and the error tolerance constant $\epsilon$. We select the coefficients such that all these three loss terms have similar scales. For the error tolerance constant $\epsilon$, we tune $\epsilon$ to deal with the noise of the preference oracle while not deteriorating the performance of POEM when the preference oracle is accurate. We show the ablation study on the hyperparameter, the error tolerance constant $\epsilon$, in Section C.7.

*Table 2.* Default Hyper-Parameters and Configurations

| Hyper-Parameter | Default Configuration |
|---|---|
| Dimension of latent embedding $d$ | 5 |
| Discount factor $\gamma$ | 0.99 |
| Optimizer (all losses) | Adam (Kingma & Ba, 2014) |
| Learning rate of policy network | $1 \cdot 10^{-4}$ |
| Learning rate of Q function | $1 \cdot 10^{-4}$ |
| Learning rate of Value function | $3 \cdot 10^{-4}$ |
| Learning rate of encoder | $1 \cdot 10^{-4}$ |
| Adam-$(\beta_1, \beta_2, \epsilon)$ | $(0.9, 0.999, 10^{-8})$ |
| $\beta_r$: coefficient of reward loss $\mathcal{L}_r$ | 0.2 |
| $\beta_{pre}$: coefficient of preference loss $\mathcal{L}_{pre}$ | 6.0 |
| $\beta_\pi$: coefficient of policy loss $\mathcal{L}_\pi$ | 0.5 |
| Error tolerance constant $\epsilon$ | 0.1 |
| # Gradient steps per environment step | 1/5 |
| # Gradient steps per target update | 1 |
| # Transitions in replay buffer (for each task $\mathcal{T}$) | 1e5 |
| # Tasks in each mini-batch for training SAC | 32 |
| # Transitions in each task batch for training SAC | 256 |
| # Trajectories in each task of each mini-batch | 10 |
| # Transitions in each context | 128 |
| # Preference queries $K$ | 10 |
| # Sample embeddings $N_z$ | 100 |
| # Trajectory horizon | 200 |

*Table 3.* Comparison of average accumulated reward $\pm$ standard deviation between POEM and baseline methods in the adaptation from human feedback (after querying 1, 5, and 10 times preference oracle, respectively) on MuJoCo and MetaWorld.

| | HalfCheetah-Vel | | | HalfCheetah-Fwd-Back | | |
|---|---|---|---|---|---|---|
| | 1 query | 5 queries | 10 queries | 1 query | 5 queries | 10 queries |
| MAML-reward | -201$\pm$13 | -193$\pm$11 | -182$\pm$11 | -120$\pm$130 | -40$\pm$138 | -4$\pm$142 |
| ANOLE | -173$\pm$12 | -120$\pm$10 | -112$\pm$14 | 503$\pm$149 | 1012$\pm$121 | 1198$\pm$89 |
| ANOLE-random | -172$\pm$14 | -125$\pm$13 | -115$\pm$13 | 1236$\pm$203 | 1419$\pm$108 | 1483$\pm$45 |
| ANOLE-greedy | -169$\pm$12 | -128$\pm$18 | -115$\pm$12 | 1367$\pm$155 | 1482$\pm$91 | 1493$\pm$89 |
| POEM | -65$\pm$10 | -50$\pm$8 | -46$\pm$6 | 1689$\pm$54 | 1684$\pm$32 | 1709$\pm$23 |

| | Ant-Goal | | | Ant-Fwd-Back | | |
|---|---|---|---|---|---|---|
| | 1 query | 5 queries | 10 queries | 1 query | 5 queries | 10 queries |
| MAML-reward | -610$\pm$81 | -603$\pm$85 | -602$\pm$89 | 101$\pm$11 | 105$\pm$12 | 105$\pm$11 |
| ANOLE | -550$\pm$21 | -405$\pm$29 | -380$\pm$19 | 618$\pm$60 | 830$\pm$20 | 802$\pm$38 |
| ANOLE-random | -551$\pm$22 | -419$\pm$26 | -378$\pm$16 | 808$\pm$38 | 810$\pm$22 | 812$\pm$29 |
| ANOLE-greedy | -560$\pm$25 | -412$\pm$38 | -401$\pm$51 | 788$\pm$39 | 806$\pm$23 | 820$\pm$32 |
| POEM | -351$\pm$9 | -203$\pm$12 | -202$\pm$7 | 1030$\pm$12 | 1013$\pm$56 | 1015$\pm$48 |

| | Ant-Dir-2D | | | Ant-Vel | | |
|---|---|---|---|---|---|---|
| | 1 query | 5 queries | 10 queries | 1 query | 5 queries | 10 queries |
| Meta-reward | 102$\pm$20 | 101$\pm$23 | 112$\pm$30 | -3$\pm$9 | -1$\pm$10 | 6$\pm$8 |
| ANOLE | 308$\pm$45 | 515$\pm$78 | 588$\pm$67 | 36$\pm$5 | 80$\pm$10 | 83$\pm$7 |
| ANOLE-random | 181$\pm$30 | 432$\pm$45 | 498$\pm$35 | 20$\pm$5 | 60$\pm$10 | 100$\pm$6 |
| ANOLE-greedy | 172$\pm$20 | 443$\pm$30 | 414$\pm$25 | 30$\pm$5 | 70$\pm$10 | 110$\pm$5 |
| POEM | 671$\pm$15 | 732$\pm$45 | 749$\pm$40 | 95$\pm$5 | 108$\pm$8 | 112$\pm$4 |

| | Humanoid-Dir-2D | | | MetaWorld-ML10 | | |
|---|---|---|---|---|---|---|
| | 1 query | 5 queries | 10 queries | 1 query | 5 queries | 10 queries |
| MAML-reward | 504$\pm$52 | 529$\pm$56 | 523$\pm$58 | 153$\pm$18 | 164$\pm$19 | 166$\pm$15 |
| ANOLE | 754$\pm$59 | 703$\pm$55 | 722$\pm$62 | 238$\pm$31 | 192$\pm$42 | 240$\pm$21 |
| ANOLE-random | 232$\pm$54 | 780$\pm$64 | 796$\pm$58 | 242$\pm$31 | 192$\pm$39 | 229$\pm$29 |
| ANOLE-greedy | 231$\pm$30 | 781$\pm$59 | 798$\pm$54 | 239$\pm$34 | 175$\pm$26 | 183$\pm$13 |
| POEM | 812$\pm$45 | 1462$\pm$14 | 1451$\pm$67 | 326$\pm$17 | 324$\pm$13 | 349$\pm$7 |

| | MetaWorld-ML1 | | |
|---|---|---|---|
| | 1 query | 5 queries | 10 queries |
| MAML-reward | 680$\pm$36 | 691$\pm$40 | 692$\pm$49 |
| ANOLE | 1086$\pm$t67 | 1042$\pm$70 | 1018$\pm$82 |
| ANOLE-random | 1117$\pm$64 | 1045$\pm$80 | 1145$\pm$52 |
| ANOLE-greedy | 1091$\pm$47 | 1011$\pm$67 | 1122$\pm$76 |
| POEM | 1435$\pm$12 | 1588$\pm$32 | 1601$\pm$13 |

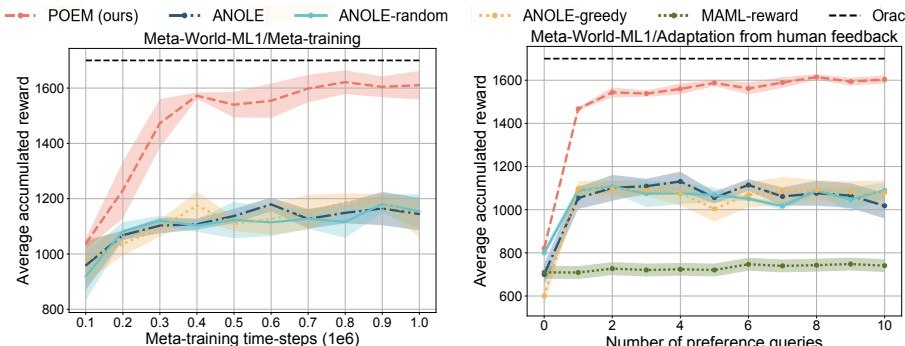

*Figure 9.* Performance on MetaWolrd-ML1. **Left:** Average accumulated reward on test tasks v.s. samples collected during meta-training. **Right:** Average accumulated reward on test tasks v.s. preference queries during adaptation from human feedback. "Oracle" denotes the performance when the reward signals are available during the meta-test.

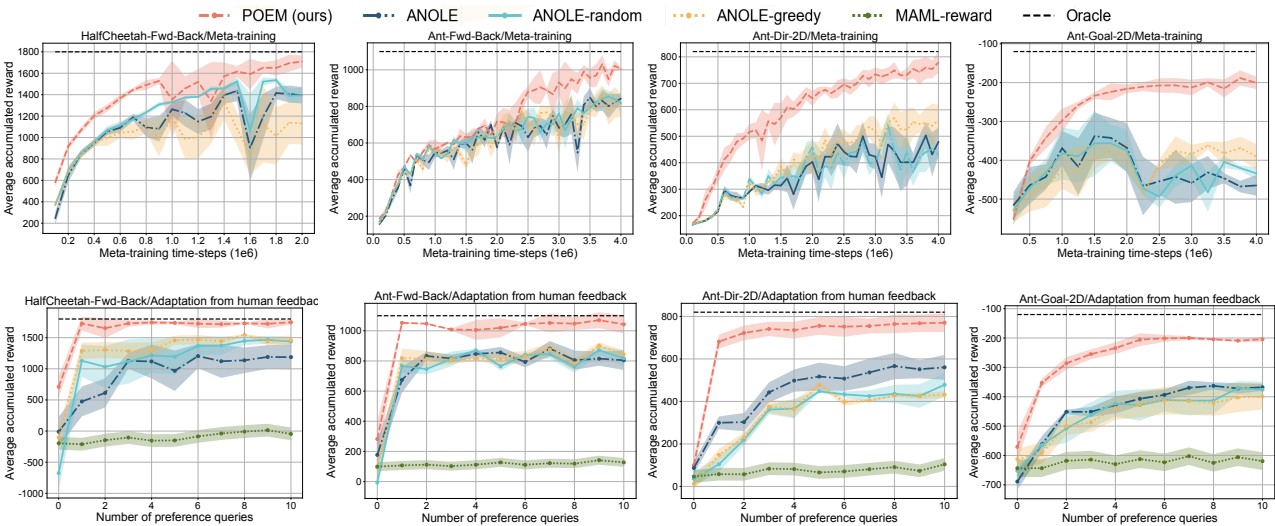

*Figure 10.* Performance on HalfCheetah-Fwd-Back, Ant-Fwd-Back, Ant-Dir-2D, and Ant-Goal. **Top:** Average accumulated reward on test tasks v.s. samples collected during meta-training. **Bottom:** Average accumulated reward on test tasks v.s. preference queries during adaptation from human feedback. "Oracle" denotes the performance when the reward signals are available during the meta-test.

### C.5. Supplemental results for performance evaluation

Figures 9 and 10 show the performance evaluation of POEM and baseline methods with an accurate preference oracle on MetaWolrd-ML1, HalfCheetah-Fwd-Back, Ant-Fwd-Back, Ant-Dir-2D, and Ant-Vel. It is shown that the proposed method, POEM, significantly outperforms all the baseline methods during both the meta-training and the meta-test.

We further compare the proposed method POEM with the baseline methods in Table 3. It is shown that the proposed method, POEM, significantly outperforms all the baseline methods. Moreover, a single query for the adaptation from human feedback can achieve the best performance on HalfCheetah-Fwd-Back and Ant-Fwd-Back, because these two scenarios are simple and each scenario only includes two tasks, and a single query is sufficient to distinguish the task embedding of a given new task. In the other seven scenarios, 5 queries for the adaptation from human feedback almost achieve the best performance. Moreover, in these harder scenarios, the proposed method, POEM, exhibits greater performance advantages over baseline methods compared to that in the two simpler scenarios.

## C.6. Supplemental results under noisy preference oracle

We evaluate POEM when the preference oracle holds noises during the adaptation from human feedback. We use the Boltzmann model to model the human preference errors. In particular, the oracle answers $\tau^{(1)} \succ \tau^{(2)}$ with the probability

$$\frac{\exp\left(\beta \cdot \text{Return}\left(\tau^{(1)}\right)\right)}{\exp\left(\beta \cdot \text{Return}\left(\tau^{(1)}\right)\right) + \exp\left(\beta \cdot \text{Return}\left(\tau^{(2)}\right)\right)} = \frac{\exp\left(\beta \sum_t r_t^{(1)}\right)}{\exp\left(\beta \sum_t r_t^{(1)}\right) + \exp\left(\beta \sum_t r_t^{(2)}\right)}$$

where $\beta$ denotes the temperature parameter. This error mode is commonly considered by recent preference-based RL works (Ren et al., 2022; Lee et al., 2021a). To normalize the total reward, we select the temperature parameter $\beta = \frac{1}{|\sum_t r_t^{(1)}| + |\sum_t r_t^{(2)}|}$.

Figure 11 illustrates the performance of POEM under a noisy preference oracle on MetaWolrd-ML1, HalfCheetah-Fwd-Back, Ant-Fwd-Back, Ant-Dir-2D, and Ant-Vel. The results indicate that POEM achieves performance comparable to that observed with an accurate preference oracle, as shown in Figures 9 and 10, highlighting its robustness to preference errors. Notably, while (Ren et al., 2022) has demonstrated ANOLE's strong performance in handling preference noises, the proposed method significantly outperforms ANOLE in these scenarios.

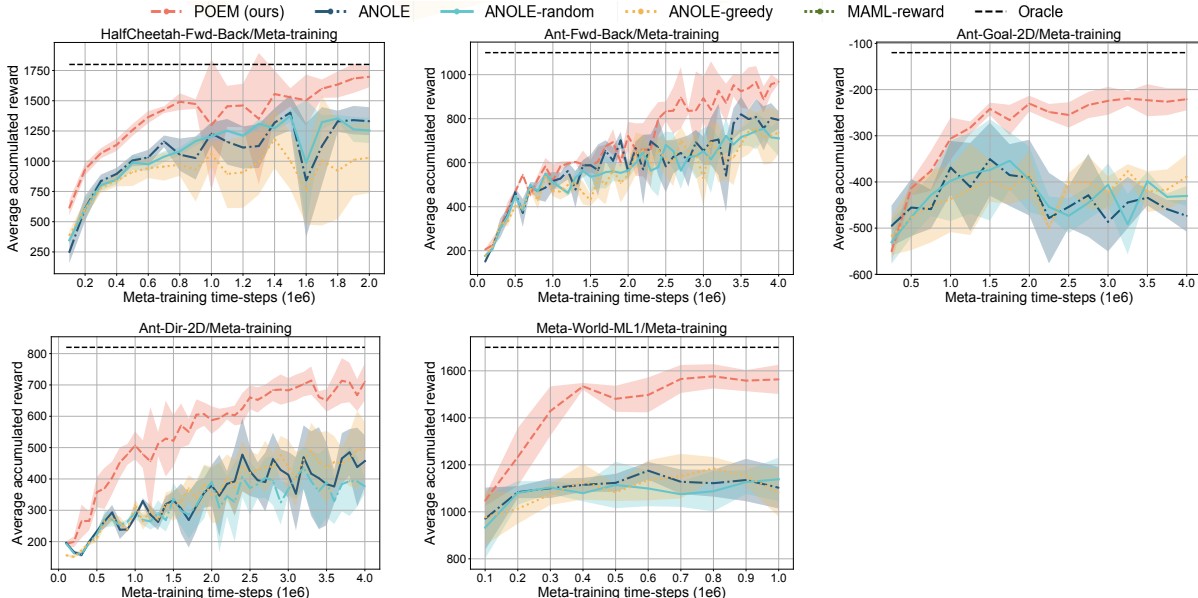

*Figure 11.* Performance evaluation under a noisy preference oracle. Average accumulated reward on test tasks v.s. samples collected during meta-training on HalfCheetah-Fwd-Back, Ant-Fwd-Back, Ant-Goal, Ant-Dir-2D, and MetaWorld-ML1.

## C.7. Supplemental results for ablation studies

We conduct ablation studies to test the impact of the individual design components in POEM, including (i) two-encoder structure, i.e., the policy/reward embedding encoders (Section 5.1) and (ii) context design (Section 5.2).

**Two-encoder structure with the new distance metric.** We first consider the impact of the two-encoder structure. In Section 5.1 and Figure 3, we employ an embedding encoder to map the task context $c^{\mathcal{T}}$ to the task embedding $z^{\mathcal{T}}$, and then employ two mappings the task embedding $z^{\mathcal{T}}$ to the task-specific policy embedding $z_\pi^{\mathcal{T}}$ (by $f_{\phi_\pi}$) and task-specific reward embedding $z_r^{\mathcal{T}}$ (by $f_{\phi_r}$). Moreover, we consider the distance metric $S(z^{\mathcal{T}_1}, z^{\mathcal{T}_2}) \triangleq \langle z_r^{\mathcal{T}_1}, z_\pi^{\mathcal{T}_2} \rangle$, where $z_\pi^{\mathcal{T}} = f_{\phi_\pi}(z^{\mathcal{T}})$ and $z_r^{\mathcal{T}} = f_{\phi_r}(z^{\mathcal{T}})$. Here, we consider a naive option to define the task embedding and the distance metric, i.e. employing a single embedding encoder to map the task context $c^{\mathcal{T}}$ to the task embedding $z^{\mathcal{T}}$ and then define the distance metric $S(z^{\mathcal{T}_1}, z^{\mathcal{T}_2}) \triangleq \langle z^{\mathcal{T}_1}, z^{\mathcal{T}_2} \rangle$. In Proposition 1 of Appendix B.1.1, we theoretically show the ineffectiveness of the naive distance metric. Here, we experimentally compare the performance between the two designs.

Figure 12 shows the performance comparison between the two-encoder structure with the distance metric in 3 designed for POEM and the single encoder with naive distance metric. It is shown that using the single encoder with naive distance metric hurts performance.

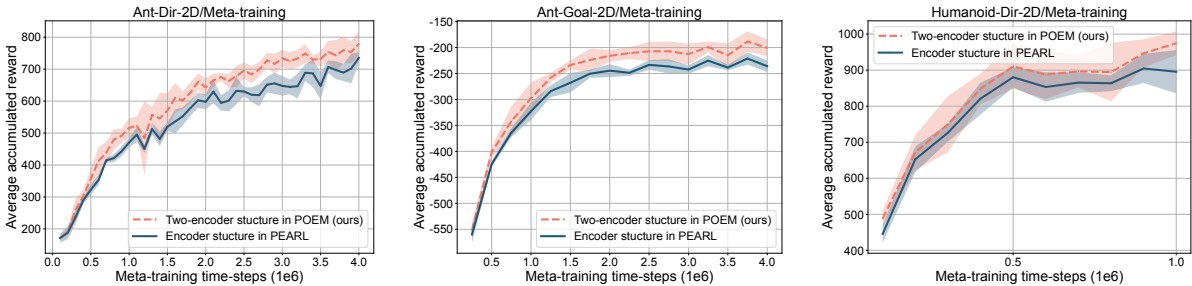

*Figure 12.* Ablation studies on the encoder structures in Ant-Dir-2D, Ant-Goal, and Humanoid-Dir. Average accumulated reward on test tasks v.s. samples collected during meta-training.

**Context design and recursive training.** We examine our choice of context, i.e., the input of the encoder. As introduced in Section 5.2, we consider the task context as $c^{\mathcal{T}} = \{(s^{(t)}, \hat{a}^{(t)}, s^{(t+1)}, r_{\mathcal{T}}^{(t)}, Q_{\mathcal{T}}^{\pi_{\mathcal{T}}^*}(s^{(t)}, \hat{a}^{(t)}))\}_{t=1}^H$, where $\hat{a}^{(t)} \sim \pi_{\mathcal{T}}^*(\cdot|s^{(t)})$ and $Q_{\mathcal{T}}^{\pi_{\mathcal{T}}^*}(s^{(t)}, \hat{a}^{(t)})$ is the optimal value on $(s^{(t)}, \hat{a}^{(t)})$. As discussed in Section 5.3, since the optimal policy $\pi_{\mathcal{T}}^*$ is not available during the meta-training, we approximate the optimal policy $\pi_{\mathcal{T}}^*$ by the policy $\pi_{\theta_{\pi}^{(n-1)}}(\cdot, z^{\mathcal{T}_{\pi},(n-1)})$, where both the policy network $\pi_{\theta_{\pi}^{(n-1)}}$ and the embedding $z^{\mathcal{T}_{\pi},(n-1)}$ are obtained from the last epoch $n-1$. Then, we use the recursive training to update the parameters of the policy network $\theta_{\pi}^{(n)}$ and the embedding $z^{\mathcal{T}_{\pi},(n)}$ in each epoch $n$.

In PEARL (Rakelly et al., 2019), the context is $c^{\mathcal{T}} = \{(s^{(t)}, a^{(t)}, s^{(t+1)}, r_{\mathcal{T}}^{(t)})\}_{t=1}^H$, where $a^{(t)}$ is sampled by posterior sampling. This enables the trajectory sampling for the new task during the meta-test. However, in the meta-RL with adaptation from human feedback, the trajectory with reward signals is not available for the new task, and the proposed method does not need to sample the trajectory for the context. Therefore, we do not need to use the posterior sampling in PEARL. Instead, we design the context which incorporates more information about the optimal policy in $\hat{a}^{(t)}$ and $Q_{\mathcal{T}}^{\pi_{\mathcal{T}}^*}$.

We experimentally verify the benefits of the context design in POEM over that in PEARL. Figure 12 shows the performance comparison between the context design in POEM and that designed for PEARL. It is shown that using the context designed for PEARL holds certain performance loss. This could be attributed to the fact that the optimal policy-related information embedded within the designed context aids the encoder in learning more informative and effective task embeddings.

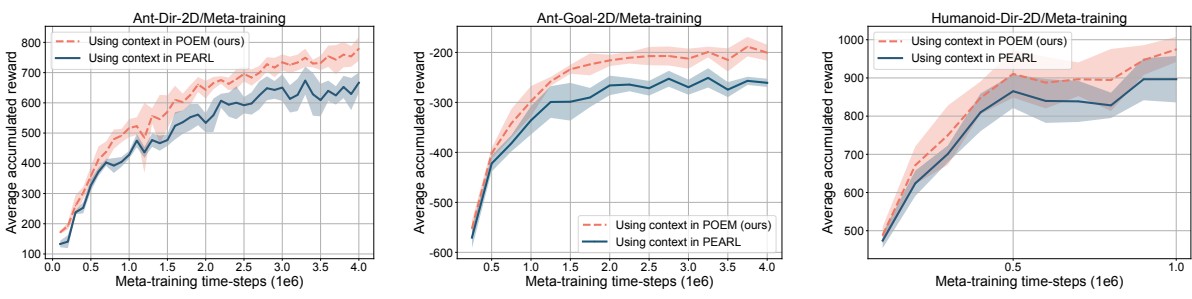

*Figure 13.* Ablation studies on the context design in Ant-Dir-2D, Ant-Goal, and Humanoid-Dir. Average accumulated reward on test tasks v.s. samples collected during meta-training.

**Ablation study on selecting the error tolerance constant $\epsilon$.** We conduct the ablation study on the hyperparameter, the error tolerance constant $\epsilon$, under the preference oracle and the preference oracle with noise set as Appendix C.6. Figure 14 shows the results of different error tolerance constants $\epsilon$ in the adaptation from human feedback (meta-test) under the preference oracle. Figure 15 shows the results of different error tolerance constants $\epsilon$ in the adaptation from human feedback (meta-test) under the noisy preference oracle.

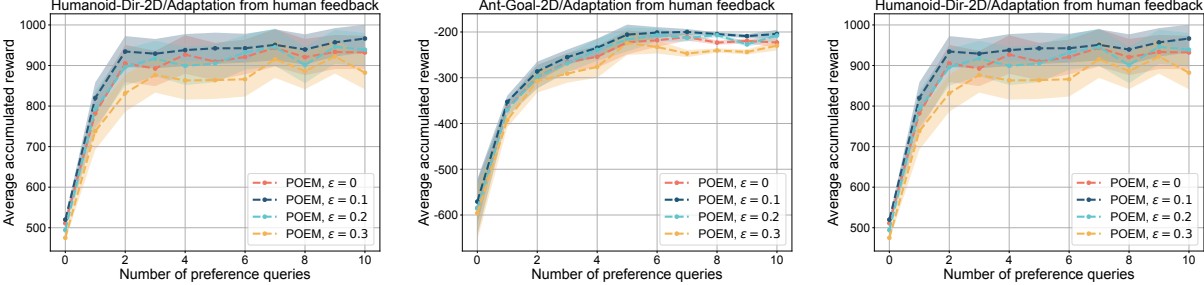

*Figure 14.* Ablation studies on selecting the error tolerance constant $\epsilon$, in Ant-Dir-2D, Ant-Goal, and Humanoid-Dir. Average accumulated reward on test tasks v.s. samples collected during meta-training. The human feedback is the oracle.

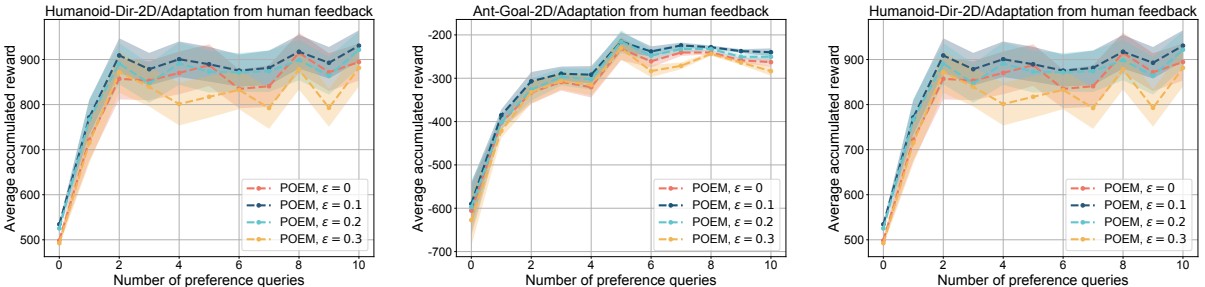

*Figure 15.* Ablation studies on selecting the error tolerance constant $\epsilon$, in Ant-Dir-2D, Ant-Goal, and Humanoid-Dir. Average accumulated reward on test tasks v.s. samples collected during meta-training. The human feedback is a noisy oracle.

