# OpenReview forum: "Meta-Reinforcement Learning with Adaptation from Human Feedback via Preference-Order-Preserving Task Embedding"
_ICML.cc/2025/Conference — ICML 2025 poster_

### Official Review · Reviewer_Fiat · 2025-03-06

**Overall Recommendation:** 3

**Summary:**

This paper focuses on meta-reinforcement learning with human in the loop adaptation scenario and proposes the Preference-Order-preserving EMbedding framework. The core idea of this framework is that if the optimal strategy of the environment achieves better performance in the other task, then the two tasks are more similar and the task embeddings are also closer. During the training phase, the framework trains an encoder and aligns the task embeddings with the preferences. In the human in the loop adaptation phase, the framework employs a task embedding inference method.

## update after rebuttal
My concern was resolved, and I raised my score.

**Claims And Evidence:**

Yes

**Essential References Not Discussed:**

No.

**Experimental Designs Or Analyses:**

Yes.

**Methods And Evaluation Criteria:**

Yes.

**Other Comments Or Suggestions:**

No.

**Other Strengths And Weaknesses:**

Strengths:1.
1.This paper focuses on meta-RL with human-in-the-loop adaptation, which I believe is an important topic.
2.The authors perform an evaluation on MuJoCo and Meta-World, demonstrate better performance compared to SOTA methods.
3.The paper is well-written, clearly explaining its contributions and well-grounded in prior literature.

Weaknesses:
The use of the policy optimzation loss function to replace the optimal policy in Algorithm 1 requires further discussion.
2.The motivation behind Algorithm 2 needs to be explained in more detail.
3.Additionally, experiments on dynamic changing tasks should be included.

**Questions For Authors:**

1.In Algorithm 1, the policy optimization loss is used to replace the optimal policy, while the preference loss requires the policy to be optimal. However, both losses are used simultaneously at the beginning of training. Would it be better to use only the policy optimization loss initially and introduce the preference loss after n epochs?
2.I still cannot fully understand why Equation (9) and line 17 of Algorithm 2 are designed in this way. Could the authors provide further explanation in plain language as to why this formulation leads to high query efficiency?
3.In Algorithm 2, would it be beneficial to further query the preferences of z in Q? For example, based on the pairs ($z_{1}$,$z_{2}$) and ($z_{3}$,$z_{4}$) in Q, could querying ($z_{1}$,$z_{3}$) lead to better results.
4.The method should also be applicable to tasks with dynamic changing. How does the method perform in such environments?
If the authors address the concerns I have raised, I'm willing to increase my score.

**Relation To Broader Scientific Literature:**

The authors propose a novel method building upon the broader scientific literature.

**Theoretical Claims:**

No.

---

> ### Author Rebuttal · Authors · 2025-04-01
>
> We are grateful and indebted for the time and effort invested in evaluating our manuscript and for all the suggestions to make our manuscript better.
>
> >**Weakness 1 and Q1**
>
> **Answer:** Thanks for pointing out the important observation. In both the existing context-based meta-RL methods, such as PEARL (Rakelly et al., 2019), and this paper, the ideal case of the meta-training can be using the optimal policy to reconstruct the decoder policy, and using the optimal policy to train the encoder. As you kindly mentioned, this can help avoid the error due to the current policy being not optimal. However, the optimal policies are not accessible, we have to replace the optimal policy reconstruction loss by the policy optimization loss.
>
> In PEARL, the authors also use the policy optimization loss to train the output conditional policy (the decoder policy) and train the encoder simultaneously at the beginning of training. Although the policies during the training for the encoder are not optimal, the method is shown to be effective and efficient. One reason could be that although the policies are not optimal, they could provide an effective approximation to guild the optimization during the training to the correct direction. Therefore, we follow the same encoder-decoder training pattern in this paper, which keeps the algorithm statement concise.
>
> >**Weakness 2 and Q2**
>
> **Answer:** Explanation of Equation (9), i.e., $(\hat{z}^{\prime}, \hat{z}^{\prime\prime})={\arg\min}_{z^{\prime},z^{\prime\prime} \in \mathcal{Z}_k}( \max \lbrace|\mathcal{Z}^{(1)}|,|\mathcal{Z}^{(2)}|  \rbrace )$:
>
> In Equation (9), $\mathcal{Z}^{(1)}=\lbrace z\in \mathcal{Z}\_k: S(z,z^{\prime} ) >\_\epsilon S(z,z^{\prime\prime} )\rbrace$ and $\mathcal{Z}^{(2)}=\lbrace z \in \mathcal{Z}\_k: S(z,z^{\prime\prime} ) >\_\epsilon S(z,z^{\prime})\rbrace$.
> When $z^{\prime}$ is preferred over $z^{\prime\prime}$, i.e., $S(z,z^{\prime} ) >\_\epsilon S(z,z^{\prime\prime} )$, any embedding in $\mathcal{Z}^{(1)}$ will satisfy the preference condition and remain to the valid embedding candidate. Similarly, when $z^{\prime\prime}$ is preferred over $z^{\prime}$, i.e., $S(z,z^{\prime\prime} ) >\_\epsilon S(z,z^{\prime} )$, any embedding in $\mathcal{Z}^{(2)}$ will remain as the valid embedding candidate. If the number of the remaining valid embedding candidates is smaller, the range of the valid embeddings will be quickly narrowed, and then the query efficiency will be higher.
> Therefore, the goal of Equation (9) is to make the number of the remaining valid candidates as small as possible.
> As we do not know what the queried preference is and which set of $\mathcal{Z}^{(1)}$ and $\mathcal{Z}^{(2)}$ will be remaining, we have to consider the worst case, i.e., the one between $\mathcal{Z}^{(1)}$ and $\mathcal{Z}^{(2)}$ with the larger size
> will be remaining. Therefore, Equation (9) first takes the set $\max \lbrace |\mathcal{Z}^{(1)}|,|\mathcal{Z}^{(2)}|\rbrace$ to pick the larger set in $\mathcal{Z}^{(1)}$ and $\mathcal{Z}^{(2)}$, and then minimizes its size.
>
> Explanation of line 17 in Algorithm 2, i.e., $z^k=
> \mathop{\arg\max}\_{z \in \mathcal{Z}_k} {\sum}\_{(z^{\prime},z^{\prime\prime}) \in \mathcal{Q}\_k} \log \mathrm{Pr} [S(z,z^{\prime} )>S(z,z^{\prime\prime})]$:
>
> In the $k$-th iteration of the human-in-the-loop adaptation, the candidate embedding set $\mathcal{Z}_k$ includes multiple valid embedding candidates. However, we need to pick one as the output for the $k$-th iteration. Therefore, line 17 is to use the maximum likelihood over all the valid embedding candidates in $\mathcal{Z}_k$ to determine the output task embedding $z^k$.
>
> >**Q3**
>
> **Answer:** After query the preferences of $(z_1,z_2)$ and $(z_3,z_4)$, it will be beneficial from querying $(z_1,z_3)$. However, $(z_1,z_3)$ is not the most query-effective pair to be queried. As mentioned in the **Answer to Q2**, we use Equation (9) to determine which pair is the most query-effective.
>
> Note that, in Algorithm 2, sampling two new embeddings other than $z_1, z_2, z_3, z_4$ in $Q$, incurs almost no cost, as it merely involves sampling from a normal distribution. However, the human preference is the most expensive step in the human-in-the-loop adaptation. Therefore, $(z_1,z_3)$ in $Q$ will not be used to query the human preference if it is not the most query-effective pair computed by Equation (9).
>
> >**Weakness 3 and Q4**
>
> **Answer:** Our method cannot handle tasks in which the dynamics change over time within a single task. If the dynamics is changing over time within any single task. When the task is revealed to the agent, the agent needs to query a human for preference comparison gradually and optimize the policy based on all the historical preference queries. However, when the dynamics change over time, the historical preference queries from humans may become invalid, which leads to a wrong direction for the policy optimization. Solving tasks with dynamics changing over time is an interesting future work.

---

### Official Review · Reviewer_oXJH · 2025-03-10

**Overall Recommendation:** 4

**Summary:**

This paper presents a novel meta-reinforcement learning (meta-RL) framework called Preference-Order-Preserving EMbedding (POEM), which enables test-time preference-based human-in-the-loop adaptation of the meta-RL policy. The main research problem is how to meta-train a policy when there exists a discrepancy between the reward feedback available at training time, and human preference-only feedback at test time. The paper proposes a preference-order-preserving task embedding algorithm that maps tasks into a latent space where task embedding similarity preserves preference orderings. At test-time, the method infers task embeddings from human preference queries and finds the best task-specific policy. Both theoretical and empirical results are presented in the paper. Theoretical results include a provably preference-order preserving similarity metric that addresses an issue with standard cosine similarity, and convergence guarantee for the adapted policy at test time. Empirical results compare POEM with baseline meta-RL methods such as ANOLE and MAML on standard simulated robotic continuous-control domains, showing significant improvement.

**Claims And Evidence:**

1. The paper presents a preference-order preserving task encoder for context-based meta-RL learning that enables human-preference adaptation at test-time.

This claim is strongly supported by the description of POEM, theoretical derivation of similarity and the resulting encoder implementation, theoretical convergence guarantee, and details on the meta-train and adaptation algorithms.

2. POEM achieves performance comparable to meta-RL oracle, with a 20%-50% improvement over baselines on standard continuous control tasks.

This claim is strongly supported by experiment results on nine continuous control environments using standard benchmarks (MuJoCo and MetaWorld) and comparisons against multiple baselines (ANOLE, MAML-reward).

**Essential References Not Discussed:**

None.

**Experimental Designs Or Analyses:**

The experimental design is well done, with ablations for different components of POEM. Details of the experiment sets, including evaluation tasks, hyperparameters, are given. Experiments also study how noise in the preference oracle affects the learned policy.

**Methods And Evaluation Criteria:**

The methods and evaluation criteria used in the paper are mostly appropriate for the problem.

One minor weakness in the evaluation is the experiments are limited to continuous control robotics tasks, and the task spaces are mostly target locations. In these setups the level of difficulty in generalizing to different goals may not be very high. How would POEM perform in more complex task spaces?

**Other Comments Or Suggestions:**

L196, right column “the embedding \tau_{\pi}” -> “the embedding \tau_{r}”

L324, left column “PREAL” -> “PEARL”

L347, right column “mile” -> “mild”

L383, right column “APACE” -> “POEM”

L760, “mile” -> “mild”

**Other Strengths And Weaknesses:**

Presentation of the paper is a clear strength. From motivation to the theoretical results to implementation choices are all clearly explained.

**Questions For Authors:**

None.

**Relation To Broader Scientific Literature:**

The paper is well-placed within existing works on context-based meta-RL.

**Theoretical Claims:**

The paper makes 2 theoretical claims, both supported by proofs.

Theorem 1: Proves that an embedding space can be constructed where task similarity aligns with preference order.
Theorem 2: Shows that the adapted task embedding converges to the true task embedding with sufficient preference queries.

---

> ### Author Rebuttal · Authors · 2025-04-01
>
> We are grateful and indebted for the time and effort invested in evaluating our manuscript. Thanks for the typo reminders and the suggestions to make our manuscript a better and stronger contribution.
>
> >**Methods And Evaluation Criteria: One minor weakness in the evaluation is the experiments are limited to continuous control robotics tasks, and the task spaces are mostly target locations. In these setups the level of difficulty in generalizing to different goals may not be very high. How would POEM perform in more complex task spaces?**
>
> **Answer:** In the conducted experiment, MetaWorld-ML10, the task is defined by the type of manipulation rather than the target location. The task family includes multiple types of manipulation tasks. As shown in Figure 8 (page 17), the interactions between the robot and the environment vary across tasks, encompassing different types of manipulation such as ‘assembly,’ ‘basketball,’ and ‘door opening.’ The training task set in this benchmark is highly heterogeneous, making the task space complex. As shown in Figure 5, the proposed method achieves a 50\% improvement over the baselines on MetaWorld-ML10.

---

> > ### Comment · Reviewer_oXJH · 2025-04-08
> >
> > I thank the authors for the rebuttal. I maintain my original score.

---

### Official Review · Reviewer_d1QQ · 2025-03-12

**Overall Recommendation:** 3

**Summary:**

The paper presents a framework for meta-reinforcement learning (meta-RL) called Preference-Order-preserving Embedding (POEM), which aims to facilitate few-shot adaptation to new tasks using human preference queries instead of traditional reward signals. The framework comprises a task encoder that maps tasks into a preference-order-preserving embedding space and a decoder that generates task-specific policies from these embeddings. During the adaptation process, the encoder efficiently infers embeddings for new tasks based on human preferences, ensuring that task embeddings reflect similarity in task performance. The authors provide theoretical guarantees for the convergence of the adaptation process to optimal task-specific policies and demonstrate through experiments that POEM significantly outperforms existing methods, achieving a 20%-50% improvement in performance on various continuous control tasks.

**Claims And Evidence:**

No. The approach is quite similar to the work proposed by ANOLE. The only difference is the partitioning of the embedding into reward and policy spaces, which seems like a relatively incremental improvement.

**Essential References Not Discussed:**

SoA is satisfactory

**Experimental Designs Or Analyses:**

The authors compute extensive results, demonstrating impressive improvements over the state-of-the-art.  However, a) it is unclear what would happen if the environment configuration changes across tasks.  b) If there is an imbalance in tasks distribution, it could affect π-θ, which in turn might introduce an inductive bias in the encoder. This issue is not addressed or discussed in the paper.

**Methods And Evaluation Criteria:**

Yes . The method tries to solve very relevant problem in RL.

**Other Comments Or Suggestions:**

Nothing

**Other Strengths And Weaknesses:**

Strength

The paper formulates the theorem and its proof in a satisfactory manner.
The authors also compute extensive results, demonstrating impressive improvements over the state-of-the-art.

Weakness
The comparison with ANOLE might not be entirely fair. In their paper, they meta-train for 4M steps on the ant task and 2M steps on the cheetah task, while this paper only trains for 1M steps. It is unclear what would happen if they trained for a similar number of steps, making direct comparisons of the results problematic.


Limited ablation study: The paper lacks an ablation study on the selection of hyperparameters.


The training details are also limited. During the meta-test phase, if multiple new tasks (T4, T5) arise, what will happen to the previously learned tasks (T1, T2, T3)? Will the learned meta-tasks perform the same way?


There are several notational and spelling errors throughout the paper, which make it difficult to understand the concepts. For instance, in lines 196-200.

**Questions For Authors:**

Refer weakness section

**Relation To Broader Scientific Literature:**

The approach of learning composite features that consist of both reward and policy in the embedding is non-trivial. The using of human feedback data to infer the task embedding instead of using encoder and context in meta-test phase is innovative

**Theoretical Claims:**

Yes. However, Theorem 2 requires the encoder-decoder network to be well-trained. However, the definition of "well-trained" is unclear, as the policy reconstruction loss is not used initially, and it is not mentioned when it starts being used. How can we ensure that the encoder-decoder is properly trained?

---

> ### Author Rebuttal · Authors · 2025-03-28
>
> >**Claims and Evidence: The approach is quite similar to the work proposed by ANOLE.**
>
> **Answer:** The partitioning of the embedding into reward embedding and policy embedding spaces is only an initial and minor design of the paper. The main contribution of the proposed method is that we train a preference-order-preserving task encoder, which establishes a connection between task embeddings and human preferences. This connection facilitates the efficient inference of task embeddings for new tasks during human-in-the-loop adaptation.
>
> In terms of the algorithm design, as you pointed out, this paper first designs an encoder that partitions the embedding space into reward embedding space and policy embedding space. Second, based on the embedding space, we prove that an encoder exists under the embedding space partition, which holds the preference-order-preserving property (Property 1 in Section 4). Third, in Section 5, we train an encoder that enforces that Property 1 holds by using the preference loss term (equation (6) in line 241), which penalizes violations of Property 1. Fourth, in Section 6, the preference-order-preserving property of the encoder enables the task embedding inference from human preferences for the human-in-the-loop adaptation.
>
> Note that all the above four steps of the algorithm design are different from ANOLE, which addresses the issue of ANOLE that the task encoder cannot capture preference-related features across tasks.
>
> >**Theoretical Claims: The definition of "well-trained" is unclear. The policy reconstruction loss is not used. How to ensure the encoder-decoder is properly trained?**
>
> **Answer:** The definition of "well-trained" is given by the three assumptions of Theorem 1. Specifically, the assumption (i) (lines 375-377, Theorem 1) states that the posterior distribution is the normal distribution; the assumption (ii) (lines 377-379) requires that Property 1 holds, i.e., the preference-order-preserving property of the encoder holds; and the assumption (iii) (lines 379-381) requires that the optimal policy is accurately reconstructed.
>
> To achieve the above three requirements, i.e., the encoder-decoder network is well-trained, we use the KL divergence loss term in line 273 to achieve assumption (i), use the preference loss in Equation (6) (line 240) to achieve assumption (ii), use the optimal reconstruction loss in Equation (5) (line 235) to achieve assumption (iii). Therefore, including the policy reconstruction loss, all the losses included in Section 4 are used to support Theorem 2.
>
> >**Experimental Designs or Analyses a)**
>
> **Answer:** In the conducted experiment, MetaWorld-ML10, the tasks family has multiple types of manipulation tasks. As shown in Figure 8 (page 17), the interactions between the robot and the environment are different for different types of manipulation tasks, and therefore their state transition functions are different and change. For example, it is easy to see that the state transition in the "assembly" task and that in the "basketball" task are different. As shown in Figure 5, the proposed method achieves 50\% improvement over baselines on MetaWorld-ML10.
>
> >**Experimental Designs or Analyses b)**
>
> **Answer:** There are several works, such as [1,2], addressing the issue of imbalance in task distribution in meta-learning/meta-RL. Their approaches are agnostic to the meta-RL methods and can also be used to address the issue in this paper. However, this paper primarily focuses on a new problem of meat-RL with human-in-the-loop adaptation. Addressing the issue of imbalance in task distribution is an interesting future work.
>
> [1] "Learning to Balance: Bayesian Meta-Learning for Imbalanced and Out-of-distribution Tasks", 2020.
>
> [2] "Improving Generalization of Meta Reinforcement Learning via Explanation", 2024.
>
> >**Weakness 1**
>
> **Answer:** The numbers of steps between this paper and ANOLE are exactly the same for the ant task and the cheetah task. In both this paper and ANOLE (Ren et al., 2022), cheetah-vel uses 1M steps (check the first figure in Figure 5 in this paper and the second figure of Figure 1 in ANOLE paper), cheetah-fwd-back uses 2M steps (check the first figure in Figure 9 in this paper and the first figure of Figure 1 in ANOLE paper). In this paper, we also use 4M steps on the ant task (check the second and third figures in Figure 9 in this paper).
>
> >**Weakness 3**
>
> **Answer:** Meta-RL with human-in-the-loop adaptation is to train a meta-model from the training tasks (T1, T2, T3), such that it can be adapted to new tasks with limited human preference queries. The learned meta-model is fixed during the meta-test phase. When a new task T4 arises, the meta-model learned from (T1, T2, T3) will be adapted to solve T4. Similarly, when a new task T5 arises, the meta-model will be adapted to solve T5. If any of the training tasks T1, T2, T3 arises in the meta-test, the meta-model is adapted to solve the task, which is easier than solving a new task T4.

---

### Official Review · Reviewer_Pj9r · 2025-03-16

**Overall Recommendation:** 4

**Summary:**

The authors present the adaption via Preference-Order-preserving EMbedding (POEM) framework. Their key insights that are if the trajectory of a task is distilled into an embedding, the similarities between tasks should be evident in these embeddings and that the optimal policy on one task should do sufficiently well on another if there is sufficient enough similarity. They leverage these properties to create an algorithm that allows a human in the loop to do a preference-based selection between two policies to progressively move closer to the already-known policy that works best for the new task. This is essential due to the new task not providing the model with an environmental reward. They further include a relaxation on their initial insight (trajectory similarity being captured in embeddings) to account for human error in the preference selection.

**Claims And Evidence:**

The authors claim a three-fold contribution in being the "first to propose the preference-order-preserving task encoder for context-based meta-RL training, which establishes a connection between task embeddings and human preferences"; experiments with their new framework conducted in a modified Mujoco and MetaWorld; and a proof for the theoretical result for their algorithm which guarantees convergence to the optimal task-specific policy.

No claims are problematic aside from the second, primarily because I am unclear how the environments were modified despite section D in the appendix.

**Essential References Not Discussed:**

I am not aware of any essential references not discussed.

**Experimental Designs Or Analyses:**

I've read through the task descriptions of how the authors evaluated their methods in the corresponding environments and do not see any issue with these tasks, nor do I feel the environment to be invalid for the proposed algorithm. The primary concern is not understanding how these environments were modified. I did not see this explicitly outlined in D1 or D.2.

**Methods And Evaluation Criteria:**

The proposed methods follow clearly from Insights 1 and 2. I could see issues with Property 1 being called into question in scenarios where similarity between task embeddings may not be sufficient to ensure policy transferability, such as long-horizon dependencies, however I believe the relaxation in equation (8), while explained to be used to account for noise in human preference selection, should also clarify this. The environments also are well-suited to testing the proposed algorithm, though I would appreciate more detail about how the human-in-the-loop was given the trajectories (ie, text description, final reward, video?)

**Other Comments Or Suggestions:**

Despite the technical strength of this paper, I take very strong issue with the related work being relegated to the appendix. Even if the authors only use half of a column to address it in the main paper, I believe it is critically important for previous and related work to be given its proper acknowledgement by the scientific community. This is the primary motivating factor behind my score.

**Other Strengths And Weaknesses:**

This is a very strong paper. Technically, outside of lacking a few details that I may have just missed, I cannot find any fault with the paper. The proposed method is very clearly motivated and the only issues I could think of regarding their methods were already addressed. They show strong results in their evaluation and justify their approach with a rigorous proof.

**Questions For Authors:**

1.) Could the authors explain how the policies were presented to the human-in-the-loop during evaluation? How many iterations of this were required per task?

2.) Could the authors please explain the difference between the base Mujoco and their modified version?

**Relation To Broader Scientific Literature:**

The key contributions of this paper are very related to the fields of RL, specifically meta-RL, continuous control and preference ordering. I believe these contributions to be relevant to the field of Machine Learning as a whole.

**Theoretical Claims:**

I have read through and do not see any issue with the proofs provided in Appendix B.

---

> ### Author Rebuttal · Authors · 2025-04-01
>
> We are grateful and indebted for the time and effort invested in evaluating our manuscript and for all the suggestions to make our manuscript a better and stronger contribution.
>
> >**Methods And Evaluation Criteria 1: I could see issues with Property 1 being called into question in scenarios where similarity between task embeddings may not be sufficient to ensure policy transferability, such as long-horizon dependencies, however I believe the relaxation in equation (8), while explained to be used to account for noise in human preference selection, should also clarify this.**
>
> **Answer:** In this manuscript, we do **not** claim that, Property 1 always holds, and the similarity between task embeddings is sufficient to ensure the policy preference, for any encoder.
>
> Instead, we claim and prove that in Section 3, under the MDP (including the case of long-horizon dependencies), **there exists** a task encoder such that Property 1 holds, i.e., there exists an encoder such that the similarity ordering of task embedding pairs is expected to align with human preference order. Next, in Section 4, we train an encoder that enforces that Property 1 holds under the encoder.
> To achieve this, we design a loss term for the encoder network, the preference loss term (equation (6) in line 241), which penalizes violations of Property 1.
> Therefore, although Property 1 may not be naturally satisfied without the training, once the encoder is trained under the supervision of the preference loss term, Property 1 holds approximately.
>
> Furthermore, as you kindly pointed out, we also incorporate the relaxation in Equation (8) to account for noise in Property 1, enhancing the algorithm’s robustness.
>
> >**Methods And Evaluation Criteria 2: I would appreciate more detail about how the human-in-the-loop was given the trajectories**
>
> **Answer:** During the meta-test (human-in-the-loop adaptation), a new task ${\mathcal{T}}_{new}$ is given. The agent explores the environment and provides two trajectories (the rewards along the trajectories are unknown) to the human. Then, the human will tell the agent which one is better, and the agent will adapt the policy according to this human feedback.
> One time of the human preference query is denoted as one iteration of the human-in-the-loop adaptation.
> The detail of the human feedback is introduced in Section 2, lines 95-109.
>
> >**Experimental Designs or Analyses: The primary concern is not understanding how these environments were modified.**
>
> >**Question 2: Could the authors please explain the difference between the base Mujoco and their modified version?**
>
> **Answer:** Thanks for pointing out the confusion. In this paper, we do not modify the environment of Mujoco. Instead, we directly use the environment in Mujoco and design the reward functions for multiple tasks for the meta-RL setting.
> The details of the reward design are shown in Appendix C.2.
> To avoid the confusion, we will modify the "Modified Mujoco" to "Mujoco".
>
> >**Suggestions: Despite the technical strength of this paper, I take a very strong issue with the related work being relegated to the appendix. Even if the authors only use half of a column to address it in the main paper, I believe it is critically important for previous and related work to be given its proper acknowledgment by the scientific community. This is the primary motivating factor behind my score.**
>
> **Answer:** Thanks for pointing it out. In the modified version, we will move the related work section in Appendix A (lines 608 to 662) to the main body of the paper.
>
> In the related work section, we comprehensively review the works related to (i) meta-RL methods, (ii) RL from human feedback (RLHF), and (iii) methods for meta-RL with human-in-the-loop adaptation, and provide details of the comparisons between the problem settings, the problem formulations, and the algorithm designs in this paper and those in the existing works. Specifically, in (i), we discuss three categories of meta-RL methods and whether they can be applied to the problem of meta-RL with human-in-the-loop adaptation. In (ii), we discuss the existing methods for RLHF and the motivation of studying meta-RL with human-in-the-loop adaptation based on RLHF. In (iii), we discuss the existing methods for meta-RL with human-in-the-loop adaptation, including ANONLE and meta-reward, and discuss the differences between their algorithm designs and this paper.
>
> >**Question 1: Could the authors explain how the policies were presented to the human-in-the-loop during evaluation? How many iterations of this were required per task?**
>
> **Answer:** Please refer to the answer to **Methods And Evaluation Criteria 2** for the explanation of the human-in-the-loop adaptation.
>
> As shown in Figures 5 and 10, we conduct at most 10 iterations for human-in-the-loop adaptation for each task, and it usually requires about 5 iterations to achieve the near-optimal performance per task.

---

### Decision · Program_Chairs · 2025-05-01

**Decision:**

Accept (poster)

**Comment:**

The paper propose, Preference-Order-preserving Embedding (POEM), a new framework for meta-reinforcement learning (meta-RL). It aims to improve few-shot adaptation to new tasks using human preference queries. In POEM, the trajectories of different tasks are encoded into an embedding, with the similarities between tasks reflected in these embeddings. Given a new task, the task encoder facilitates efficient task embedding inference for new tasks from the preference queries and then obtains the task specific policy.

Reviewers generally agree that this paper has strong experimental results and the author has address most of the reviewers concern.